# *CodeChemist*: TEST-TIME SCALING FOR LOW-RESOURCE CODE GENERATION VIA FUNCTIONAL KNOWLEDGE TRANSFER

## ABSTRACT

Code Large Language Models (CodeLLMs) have been widely adopted for code generation, powering applications with large user bases. Their performance, however, varies sharply across programming languages (PLs) and is particularly suboptimal for low-resource PLs due to data scarcity, limiting their overall usability. In this work, we introduce *CodeChemist*, a simple yet effective test-time scaling framework that transfers the model's functional knowledge from high-resource to low-resource PLs via synthesized test cases. Specifically, *CodeChemist* first performs code generation and execution in high-resource PLs to derive test cases that capture functional knowledge, then applies multi-temperature hedged sampling to produce candidate code snippets in the low-resource PL, and finally selects the best candidate by executing the synthesized test cases. Extensive experiments demonstrate that *CodeChemist* significantly outperforms existing test-time scaling methods, improving code generation for low-resource PLs without retraining.

## 1 INTRODUCTION

Large Language Models (LLMs) have catalyzed a transformative shift in code generation, driven by the emergence of specialized variants designed for programming tasks, referred to as Code Large Language Models (CodeLLMs). With powerful capabilities in code generation, these models have consistently outperformed traditional methods and are now extensively adopted in both academic and industrial settings (Hou et al., 2024; Hui et al., 2024; Wang et al., 2025a). For example, widely used tools such as GitHub Copilot (git, 2023), which leverage models like GPT-4 and Codex (Chen et al., 2021), have greatly enhanced development efficiency through highly accurate and context-aware code generation.

However, the performance of CodeLLMs in code generation varies significantly across programming languages (PLs). They excel in high-resource PLs like Python but underperform in low-resource PLs (e.g., Lua) or those with complex syntax (e.g., C++ and Java) (Zhang et al., 2024; Giagnorio et al., 2025; Cassano et al., 2024; Tarassow, 2023). This disparity limits the practical usability of CodeLLMs in multilingual development environments and hinders support for developers using less-represented PLs (Zheng et al., 2023b). Bridging this performance gap is essential to fully realize the potential of LLMs in real-world code generation applications.

The most straightforward way to improve performance in low-resource PLs is to collect additional training data and fine-tune the model. Considering the inherent data scarcity, several lines of research have turned to cross-lingual transfer techniques that leverage corpora from high-resource PLs. For instance, Roziere et al. (2022); Cassano et al. (2024) propose translating code snippets from high-resource into low-resource PLs. In practice, translated code snippets often suffer from limited quality, and the required training process is computationally expensive. As a result, the practicality of such methods is substantially constrained.

Recently, test-time scaling methods (Li et al., 2025a) have emerged as a promising alternative to costly training-based techniques for generally enhancing code generation. However, their effectiveness in low-resource PLs is limited, as they do not consider the inherent challenge of data scarcity. Furthermore, common enhancement strategies like data augmentation are fundamentally incompatible with the test-time paradigm, making improvements for low-resource PLs particularly difficult.

Consequently, we pursue a test-time strategy that transfers the model's inherent knowledge from high-resource PLs to improve performance in low-resource PLs.

In our paper, we propose *CodeChemist*, a simple yet effective test-time scaling framework that enhances low-resource code generation by transferring functional knowledge from high-resource PLs. Its key insight is that test cases naturally encapsulate functional knowledge, which is the input-output-defined, PL-agnostic essence of a function's logic. Thus, test cases themselves serve as a novel and powerful medium for transfer at test time. In particular, the method operates through three stages. First, we generate code for a given task in a high-resource PL and execute it to derive test oracles, which are 'ground-truth' input-output pairs that encapsulate the desired functional knowledge. Next, for the low-resource PL, we employ a multi-temperature hedging strategy to produce a diverse set of code candidates. Finally, the teacher-derived test cases are used to evaluate and select the candidate whose execution behavior best matches the transferred functional knowledge.

We first conduct comprehensive experiments on Lua, a representative low-resource PL, across multiple models. The results show that *CodeChemist* achieves improvements of up to 69.5%. To further validate the extensibility of our method, we evaluate it on PLs that are considerably less low-resource, namely C++ and Java. Experimental results show that *CodeChemist* consistently improves performance across different PLs and models.

## 2 RELATED WORK

### 2.1 ENHANCING CODELLMS FOR LOW-RESOURCE PLS

CodeLLMs exhibit a significant performance gap between high-resource PLs (e.g., Python) and low-resource PLs, which has attracted considerable research attention. Existing approaches can be broadly divided into two categories: fine-tuning methods and inference-based methods.

These fine-tuning methods are typically designed to curate additional data for low-resource PLs, which is then used to fine-tune a model and enhance its performance on them. Chen et al. (2022b) propose selecting high-resource PLs for auxiliary training based on their similarity to a target low-resource PLs. For instance, to improve performance on Lua, their method prioritizes Python code for training due to its syntactic and semantic similarity. A key limitation of this approach, however, is its high task-sensitivity and limited generalization. Another line of work follows a "translation–testing–filtering" paradigm. For instance, TransCoder-ST (Roziere et al., 2022) first translates code from a high-resource PL into a low-resource PL. It then constructs a fine-tuning dataset by filtering the translated samples for validity using automatically generated unit tests. However, generating these unit tests depends on language-specific toolchains. Since many low-resource PLs lack such toolchains, this approach is difficult to generalize. MultiPL-T (Cassano et al., 2024) improves upon this by generating unit tests through CodeLLMs only in high-resource PLs. It then translates both the code and its corresponding tests into the target low-resource PLs, using execution-based verification to build a reliable training dataset. However, its effectiveness is highly dependent on the quality of the LLM-based translation for both the code and the test cases. Even with high-quality synthetic datasets, these fine-tuning-based methods can impair the model's performance on high-resource PLs. Furthermore, mastering complex linguistic constructs remains challenging even with additional targeted low-resource data.

In contrast, inference-based methods do not rely on additional training but instead exploit the intrinsic capabilities of LLMs. For example, Bridge-C (Zhang et al., 2024) first generates code with natural language annotations in a high-resource PL to serve as a reference. This annotated code is then provided as context in a prompt to guide the model in generating implementations in the target low-resource PL. However, this prompt-based approach can only produce code that mimics the provided examples and struggles with complexity, making its effectiveness contingent on the quality of the reference code.

Different from the prior work, in this paper, we propose *CodeChemist* that transfers knowledge across PLs at inference time. This approach requires no extra training data and achieves higher performance through test case validation.

## 2.2 Test Time Scaling

Test-time scaling is a technique used to enhance the reasoning capabilities of LLMs during inference by allocating more computational resources. A widely used approach is to generate multiple candidate solutions and apply a selection mechanism to choose the most promising one, commonly known as Best-of-N sampling. Within this framework, common selection strategies include (weighted) majority voting (Wang et al., 2023a), automated judgment by an LLM (LLM Judge) (Wang et al., 2025c), and scoring with a trained reward model (Christiano et al., 2017; Lightman et al., 2023). However, these strategies often struggle to identify the truly best candidate (Stroebl et al., 2024; Brown et al., 2024; Hassid et al., 2024).

Test-time scaling has also shown great potential in enhancing code generation. CodeMonkeys (Ehrlich et al., 2025) is an approach that enhances the performance of LLMs in the SWE-bench benchmark by extending test-time compute. The system generates test scripts and uses execution feedback to continuously optimize candidate code snippets. After several iterations, it combines majority voting and model selection to choose the best solution. S* (Li et al., 2025a) is a hybrid test-time extension method that uses an external model to generate test inputs and then feeds execution feedback to the LLM for optimal selection. However, the application of these methods to low-resource PLs has been largely underexplored.

## 2.3 Enhancing Code Generation through Test Cases

Using synthetic test cases to guide code generation has emerged as an effective approach (Chen et al., 2022a; Huang et al., 2024; Jiao et al., 2025). Lee et al. (2025) proposes an adversarial reinforcement learning framework that optimizes the test case generator and code generator through adversarial training, selecting the optimal code based on the number of test cases it passes. Similarly, Zeng et al. (2025) trains a reward model by constructing a problem-test case dataset and then scores the candidate code snippet to select the optimal solution. GenX (Wang et al., 2024) jointly trains the code generation model and the test generation model through execution feedback, allowing them to improve each other over time. Epicoder (Wang et al., 2025b) introduces a data synthesis framework based on feature trees and relies on LLM-generated test files to iteratively debug code.

However, the above methods rely on the model to directly generate input-output pairs, but due to hallucinations, the model may introduce inaccuracies in predicting the correct outputs.

## 3 Methodology

To transfer functional knowledge from high-resource PLs to low-resource PLs, we propose *Code-Chemist*. As shown in Figure 1, *CodeChemist* consists of three main stages: test case generation, hedged sampling, and execution-based selection. In the test case generation stage, we extract functional knowledge from high-resource PL code snippets and transfer it into PL-agnostic test cases. In the hedged sampling stage, we apply a multi-temperature hedging strategy to generate a diverse pool of candidate low-resource PL code snippets. In the execution-based selection stage, we execute the candidate code snippets on the synthesized test cases and choose the code with the highest pass rate as the final output.

### 3.1 Test Case Generation

Our test cases are input/output (I/O) pairs. Since correct code produces identical output for a given input in any PL, these I/O pairs serve as a PL-agnostic "transfer medium". To generate them, we first prompt a model to generate code in a high-resource PL. Given a programming problem $Q$ and a model $M$, we prompt the model to generate $h$ high-resource PL code snippet candidates.

Subsequently, we construct an input set that covers various scenarios. Given an initial temperature $\tau$, we provide the programming problem $Q$ to the model $M$ using prompts (see Appendix F) that encourage the generation of inputs covering both common cases and boundary scenarios (such as empty inputs), thereby capturing a more comprehensive range of behavioral knowledge.

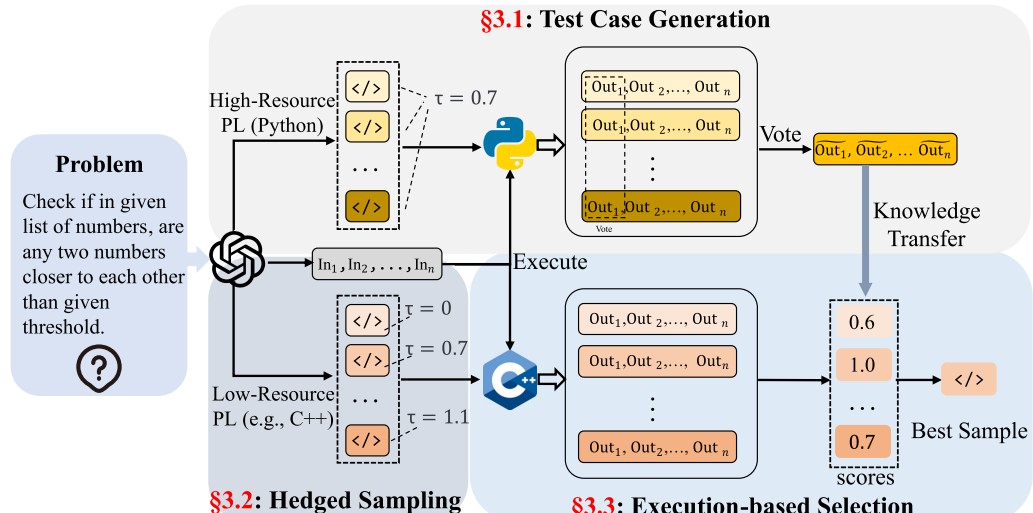

Figure 1: The overview of *CodeChemist*

Next, we execute these inputs in high-resource PLs and capture the corresponding outputs. For each input, if the program executes successfully and produces valid output, it is marked as "valid". If it encounters compilation failure, timeout, or crash, it is marked as "invalid". We collect all "valid" outputs corresponding to each input and determine the most frequent one as the final output. Thus, the final output and its corresponding test input together form a valid test oracle. In rare cases, if multiple outputs tie for the highest frequency, the test input is deemed non-discriminative and discarded. However, a single round of generation may not yield a sufficient number of valid test cases. Therefore, the model attempts to generate inputs multiple times. Specifically, after each attempt, we slightly increase the temperature parameter ($\tau + 1$) to encourage greater diversity in subsequent attempts. The generation process stops once $n$ valid input–output pairs have been collected or the maximum number of attempts is reached.

Ultimately, all I/O pairs filtered through consistency form the $n$ test cases. These test cases are semantically independent of the PL while carrying the behavioral knowledge of high-resource PLs, thereby effectively transferring this knowledge to low-resource PLs.

## 3.2 HEDGED SAMPLING

The sampling stage aims to produce a pool of candidate code snippets in the low-resource PLs that balances quality with diversity, thereby ensuring a sufficiently rich solution space for the subsequent selection stage. The key challenge lies in temperature configuration, as it directly controls the diversity-quality trade-off and must be carefully calibrated.

In standard sampling, the temperature parameter $\tau$ controls the smoothness of the softmax distribution, thereby influencing the diversity and determinism of the generated samples. For a given temperature $\tau_j$, the probability of selecting token $v_k$ is:

$$P_{\tau_j}(v_k) = \frac{\exp(l_k/\tau_j)}{\sum_i \exp(l_i/\tau_j)}.$$

$\tau$ regulates the trade-off between diversity and quality (Ye et al., 2025). When $\tau$ is large, the generated samples become more diverse. As $\tau \to 0$, the distribution sharpens and the results become deterministic. At $\tau = 0$, it corresponds to greedy decoding.

Configuring the temperature parameter $\tau$ for low-resource PLs is challenging due to two primary factors. ❶ **Inherent Uncertainty of Low-Resource PLs.** Due to limited and often lower-quality training data, low-resource PLs tend to produce "flat and uncertain" output distributions, in contrast

to the confident predictions typical of high-resource PLs. ❷ **Context-Dependent Optimality.** The optimal $\tau$ is highly context-dependent, varying significantly across models, tasks, and languages since each occupies distinct subspaces of the training distribution (Li et al., 2025b). This results in a combinatorial explosion over the combinations of model, dataset, and language, making fine-grained $\tau$ tuning prohibitively expensive and impractical for real-world applications.

Based on the above considerations, and motivated by the language-agnostic benefits of diversified sampling (Khairi et al., 2025), we adopt a multi-temperature hedged sampling strategy to generate a candidate pool of low-resource program code. This method is designed to be universally applicable across PLs, balancing quality and diversity. Specifically, we draw samples using multiple high-temperature values (to encourage diversity) while also including the greedy-decoding sample ($\tau = 0$). For instance, we selected temperatures of 0, 0.7, 0.9, and 1.1, with the number of samples being 1, 3, 3, and 3, respectively. The approach mitigates the instability typical of high-temperature sampling: the greedy sample serves as a reliable fallback when high-variance samples introduce errors, thus maintaining a baseline level of executable candidates. Meanwhile, the high-temperature variants promote diversity in structure and logic, enhancing exploration of the output space in a language-independent manner.

### 3.3 Execution-based Selection

The core of the selection stage is to use synthesized test cases to transfer functional knowledge from high-resource PLs to low-resource PLs. In the Best-of-N framework, the evaluation of candidate samples is based on an external utility function $U(y)$:

$$\hat{y} = \arg\max_{y \in Y} U(y).$$

In our approach, $U(y)$ corresponds to the execution results of the test cases obtained through knowledge transfer. Specifically, we input the test cases one by one into the low-resource PL candidate code snippets for execution. If the output matches the oracle, it will be marked as "pass", and if there is a compilation failure, timeout, or output error, it will be marked as "fail". Therefore, the pass rate of the candidate code snippets across the entire test case set becomes its utility score. Let $\mathcal{T} = \{t_1, t_2, \ldots, t_m\}$ be the set of test cases, and $y$ be the Low-resource PL candidate code snippet. The utility score $U(y)$ of $y$ is calculated as the pass rate across all test cases:

$$U(y) = \frac{1}{m} \sum_{i=1}^{m} \text{pass}(y, t_i),$$

where $\text{pass}(y, t_i) = 1$ if the candidate code snippet $y$ produces the correct output on test case $t_i$, and $\text{pass}(y, t_i) = 0$ otherwise. The candidate with the highest pass rate is selected as the final output. If all candidates receive a score of zero, we revert to the greedy ($\tau = 0$) sample. When multiple candidates attain the highest score, we prioritize programs sampled under a lower temperature. Through this mechanism, test cases serve as a medium for selecting high-quality, low-resource PL code snippets, effectively enabling cross-lingual functional knowledge transfer.

The algorithm 1 implements the *CodeChemist* framework in three sequential stages. The process begins with the stage of test case generation (Lines 1-9) that leverages a high-resource PL to produce reference implementations and test inputs, executing them to establish expected outputs through majority voting. This is followed by the multi-temperature hedged sampling (Line 10), where diverse candidate code snippets are generated in the target low-resource PL using sampling at multiple temperatures. The final stage (i.e., execution-based selection) in Lines 12 to 21 evaluates each candidate against the expected outputs from the first stage, scoring them based on functional consistency and selecting the highest-performing candidate as the final solution.

## 4 Experiments

In this section, we conduct comprehensive experiments to evaluate the effectiveness of *CodeChemist* across multiple model families, different model sizes, and various benchmark difficulties.

**Algorithm 1:** The Implementation of *CodeChemist*

**Input** : Problem $P$
**Output:** Best sample $x^*$

1   $H \leftarrow \text{GenHighCode}(P)$ ;       // Generate high-resource reference code
2   $I \leftarrow \text{GenTests}(P)$ ;              // Generate test cases
3   $O \leftarrow []$ ;             // Initialize expected outputs
4   **for** $i_j \in I$ **do**
5     $R \leftarrow \{\text{Run}(h, i_j) \mid h \in H, \text{Valid}(h, i_j)\}$ ;    // Execute high-resource codes
6     **if** $R \neq \emptyset$ **then**
7       $O \leftarrow O \cup \{\text{MajorityVote}(R)\}$ ;      // Store consensus output
8     **else**
9       $O \leftarrow O \cup \{\text{null}\}$ ;         // Mark invalid test

10   $X \leftarrow \text{MultiTempSampling}(P, \{\tau_1, \tau_2, \ldots, \tau_k\})$ ;     // Hedged sampling
11   $S \leftarrow [0] \times |X|$ ;        // Initialize score array
12   **for** $x_k \in X$ **do**
13     $p \leftarrow 0, v \leftarrow 0$ ;         // Reset counters
14     **for** $j \mid O[j] \neq null$ **do**
15       **if** $Run(x_k, I[j]) = O[j]$ **then**
16         $p \leftarrow p + 1$ ;         // Count passes
17       $v \leftarrow v + 1$ ;        // Count valid tests
18     **if** $v > 0$ **then**
19       $S[k] \leftarrow p/v$ ;        // Calculate score
20     **else**
21       $S[k] \leftarrow 0$

22   $j^* \leftarrow \arg\max S$ ;        // Find best candidate
23   **return** $x_{j^*}$ ;        // Return best sample

## 4.1 EXPERIMENT SETUP

The experimental setup includes metrics, model selection, benchmarks, comparative baselines, and implementation details.

**Metrics.** We use the Pass@1 metric to evaluate the effectiveness of code generation. To compute a robust and unbiased estimate of Pass@1, we follow the methodology introduced by Chen et al. (2021), which involves generating n=10 independent samples per problem.

**Models.** To comprehensively evaluate the performance of *CodeChemist* across models of different sizes, we select multiple variants from the same model series. Specifically, we choose the Qwen2.5-Coder-Instruct (Hui et al., 2024) (referred to as Qwen) series (including the 1.5B, 3B, 7B, 14B, and 32B versions), Llama3.2 (Dubey et al., 2024) (3B version), GPT-4o mini (Hurst et al., 2024) (referred to as 4o-mini), and introduce the DeepSeek-V3.1-chat (Liu et al., 2024) (referred to as DeepSeek) model for comparison.

**Benchmarks.** We use MultiPL-E (Cassano et al., 2022) and Ag-LiveCodeBench-X (Boruch-Gruszecki et al., 2025) as evaluation benchmarks. MultiPL-E translates HumanEval (Chen et al., 2021) and MBPP (Austin et al., 2021) into over 19 languages, with MultiPL-HumanEval retaining 161 problems from the original set and MultiPL-MBPP retaining 396 problems. Ag-LiveCodeBench-X, derived from LiveCodeBench 5.0 (Jain et al., 2024), contains 499 problems and has a higher difficulty than MultiPL-E. We evaluate the low-resource PL Lua, along with relatively low-resource PL C++ and Java.

**Baselines.** We conduct comparative experiments on MultiPL-HumanEval. First, we evaluate the performance improvement of *CodeChemist* compared to the original model (without test time scaling). Then, under the same experimental setup, we compare *CodeChemist* with several representative test-time scaling strategies, including Majority Voting (Wang et al., 2023b), LLM Judge (Zheng et al., 2023a) (using 4o-mini as the judge model), and S* (Li et al., 2025a).

**Implementation Details.** We employ a multi-temperature hedged sampling strategy to generate 10 candidate solutions for each low-resource PL problem. Specifically, the temperature is set to $t \in \{0.0, 0.7, 0.9, 1.1\}$, and 1, 3, 3, and 3 candidates are sampled in parallel, respectively. The initial temperature for test case generation is set to 0.5. Inference for the Qwen series and Llama3.2 is conducted locally on a single A100 GPU using the SGLang framework (Zheng et al., 2024), while 4o-mini and DeepSeek are accessed via their official APIs. All experimental code execution is performed on two Intel Xeon Platinum 8163 CPUs, featuring 48 cores in total with hyper-threading enabled. All inference is performed using top-p=0.95, with the specific prompt details provided in the Appendix F.

## 4.2 EXPERIMENT RESULTS

Table 1 reports the comparison of *CodeChemist* on MultiPL-HumanEval against several methods: Vanilla (no test-time scaling), Majority Voting, LLM Judge, and S*. The results demonstrate that *CodeChemist* consistently outperforms the baselines across most PLs and models, substantially enhancing the performance of low-resource PLs. Moreover, the larger the initial gap to high-resource PLs, the more pronounced the performance improvement in the low-resource PL. For example, on Qwen1.5B, the Python (63.9) vs. Lua (34.1) gap is close to 30.0 (see Appendix D), and *CodeChemist* achieves a 69.5% improvement on Lua compared with Vanilla.

Across different target PLs, *CodeChemist* demonstrates the most pronounced improvements on the low-resource PL Lua, with relative gains ranging from 5.9% to 80.6%. For the C++ language, the improvements fall within 2.2%-51.7%, while for the Java language they lie within 4.9%-60.0%. Overall, *CodeChemist* consistently improves performance across all PLs, with particularly notable gains when the performance gap between PLs is larger. This trend highlights the extensibility of our method: it is effective not only on a typical low-resource PL like Lua, but also on comparatively less low-resource PLs such as Java and C++.

Across different model families, *CodeChemist* delivers consistent performance gains across, with the effect varying by model scale and PL disparity. For smaller models, where the performance gap between high- and low-resource languages is more pronounced, *CodeChemist* achieves the most significant improvements, thereby substantially enhancing their usability on low-resource PLs. For example, on Qwen1.5B, the gains reach 69.5% for Lua, 51.7% for C++, and 60.0% for Java. For GPT-4o mini, although the performance across PLs is relatively close and the benefit from knowledge transfer is limited, *CodeChemist* still delivers gains of 7.1%, 5.5%, and 7.9%, effectively reducing the performance gap between high- and low-resource languages. For the strongest model, DeepSeek, performance across PLs is already relatively high, leaving limited room for further improvement. Nevertheless, *CodeChemist* still yields relative gains of 7.4%, 2.2%, and 4.9% on Lua, C++, and Java, respectively. This indicates that even in state-of-the-art models, cross-language knowledge transfer can play a complementary role, demonstrating the generality and robustness of the proposed method.

We showcase that our results are statistically significant via a t-test. More details are in Appendix B.

## 4.3 RESULTS ON OTHER BENCHMARK

We further evaluate *CodeChemist* on MultiPL-MBPP and Ag-LiveCodeBench-X, with the results shown in Table 2 and Table 3. On these two benchmarks, we only compare against the Vanilla methods. The experiments again demonstrate that *CodeChemist* effectively reduces the performance gap between high- and low-resource PLs, with larger gaps leading to greater gains, as observed on Qwen1.5B, and Llama3 3B.

On the relatively easy benchmark MultiPL-MBPP, *CodeChemist* achieves consistent gains, particularly on smaller models. For example, on Qwen1.5B, Lua/Java/C++ improve by 56.9%/29.3%/43.7%, respectively; as model size increases and the language gap narrows, the gains diminish accordingly (e.g., 4o-mini).

Compared with MultiPL-MBPP, Ag-LiveCodeBench-X is more difficult and closer to real-world scenarios. On this benchmark, the baseline performance of Lua is relatively poor (2.8-36.5), while *CodeChemist* achieves relative improvements ranging from 18.0% to 200.0%, effectively enhancing

Table 1: Pass@1 of Vanilla, majority voting, LLM judge, S*, and *CodeChemist* on MultiPL-HumanEval. The best performance is highlighted in bold, while the second best is underlined. Green arrows and values indicate improvements over the vanilla baseline, while red arrows and values denote a decrease in performance. For clarity and consistency in tables, we use the abbreviation Maj Voting for Majority Voting.

| Language | Method | Qwen 2.5 Coder Instruct | | | | Llama3 3B | GPT 4o -mini | DeepSeek V3.1 |
|---|---|---|---|---|---|---|---|---|
| | | 1.5B | 7B | 14B | 32B | | | |
| Lua | Vanilla | 34.1 | 69.7 | 74.2 | 78.0 | 29.9 | 74.8 | 82.1 |
| | Maj Voting | 45.3 ↑ 11.2 | 75.2 ↑ 5.5 | 77.0 ↑ 2.8 | 81.4 ↑ 3.4 | 46.6 ↑ 16.7 | 76.4 ↑ 1.6 | **88.2** ↑ **6.1** |
| | LLM Judge | 51.6 ↑ 17.5 | 73.9 ↑ 4.2 | 76.4 ↑ 2.2 | 77.0 ↓ 1.0 | 44.1 ↑ 14.2 | 75.2 ↑ 0.4 | 85.1 ↑ 3.0 |
| | S* | 50.9 ↑ 16.8 | 79.5 ↑ 9.8 | 75.8 ↑ 1.6 | 80.1 ↑ 2.1 | 50.9 ↑ 21.0 | 78.9 ↑ 4.1 | 82.0 ↓ 0.1 |
| | Ours | **57.8** ↑ **23.7** | **82.0** ↑ **12.3** | 80.8 ↑ 6.6 | 82.6 ↑ 4.6 | 54.0 ↑ 24.1 | 80.1 ↑ 5.3 | **88.2** ↑ **6.1** |
| C++ | Vanilla | 34.4 | 72.98 | 77.5 | 83.9 | 37.5 | 80.1 | 93.0 |
| | Maj Voting | 49.1 ↑ 14.7 | 79.5 ↑ 6.5 | 82.6 ↑ 5.1 | 85.7 ↑ 1.8 | 52.8 ↑ 15.3 | 83.2 ↑ 3.1 | 92.6 ↓ 0.4 |
| | LLM Judge | 45.3 ↑ 10.9 | 80.1 ↑ 7.1 | 82.0 ↑ 4.5 | 85.7 ↑ 1.8 | 51.6 ↑ 14.1 | 80.8 ↑ 0.7 | 92.6 ↓ 0.4 |
| | S* | 46.0 ↑ 11.6 | 76.4 ↑ 3.4 | 82.0 ↑ 4.5 | 85.7 ↑ 1.8 | 53.4 ↑ 15.9 | 80.1 ↑ 0.0 | 93.2 ↑ 0.2 |
| | Ours | **52.2** ↑ **17.8** | **82.6** ↑ **9.6** | **85.7** ↑ **8.2** | **87.0** ↑ **3.1** | 54.0 ↑ 16.5 | **84.5** ↑ **4.4** | **95.0** ↑ **2.0** |
| Java | Vanilla | 43.5 | 77.7 | 81.5 | 81.5 | 37.9 | 79.6 | 89.3 |
| | Maj Voting | 62.7 ↑ 19.2 | 84.8 ↑ 7.1 | 83.5 ↑ 2.0 | 83.5 ↑ 2.0 | 53.8 ↑ 15.9 | 81.7 ↑ 2.1 | 91.8 ↑ 2.5 |
| | LLM Judge | 47.5 ↑ 4.0 | 79.1 ↑ 1.4 | 80.4 ↓ 1.1 | 84.2 ↑ 2.7 | 55.7 ↑ 17.8 | 79.1 ↓ 0.5 | 93.0 ↑ 3.7 |
| | S* | 67.1 ↑ 23.6 | 84.2 ↑ 6.5 | 82.3 ↑ 0.8 | 84.2 ↑ 2.7 | 59.5 ↑ 21.6 | 84.2 ↑ 4.6 | 90.5 ↑ 1.2 |
| | Ours | **69.6** ↑ **26.1** | **85.4** ↑ **7.7** | **86.7** ↑ **5.2** | **88.6** ↑ **7.1** | 58.9 ↑ 21.0 | **86.1** ↑ **6.3** | 93.7 ↑ 4.4 |

Table 2: Pass@1 results on MultiPL-MBPP. Green arrows and values indicate improvements over the vanilla baseline, while red arrows and values denote a decrease in performance.

| Language | Method | Qwen 2.5 Coder Instruct | | | | Llama3 3B | GPT 4o -mini | DeepSeek V3.1 |
|---|---|---|---|---|---|---|---|---|
| | | 1.5B | 7B | 14B | 32B | | | |
| Lua | Vanilla | 36.9 | 57.4 | 61.3 | 61.0 | 29.6 | 62.5 | 57.8 |
| | Ours | 57.9 ↑ 21.0 | 69.8 ↑ 12.4 | 71.5 ↑ 10.2 | 66.5 ↑ 5.5 | 49.4 ↑ 19.8 | 71.0 ↑ 8.5 | 65.5 ↑ 7.7 |
| Java | Vanilla | 44.7 | 54.2 | 64.3 | 64.5 | 35.6 | 66.1 | 66.3 |
| | Ours | 57.8 ↑ 13.1 | 70.2 ↑ 16.0 | 71.0 ↑ 6.7 | 67.9 ↑ 3.4 | 51.5 ↑ 15.9 | 72.0 ↑ 5.9 | 71.2 ↑ 4.9 |
| C++ | Vanilla | 39.6 | 63.7 | 66.4 | 65.8 | 36.6 | 66.4 | 62.6 |
| | Ours | 56.9 ↑ 17.3 | 68.0 ↑ 4.3 | 71.3 ↑ 4.9 | 68.5 ↑ 2.7 | 51.1 ↑ 14.5 | 69.5 ↑ 3.1 | 71.0 ↑ 8.4 |

the performance of low-resource PLs. For C++ and Java, *CodeChemist* also provides consistent gains, with improvements of 7.3%-55.8% and 5.9%-107.1%, respectively, indicating its effectiveness even in tasks with higher algorithmic complexity and difficulty.

## 4.4 ABLATION STUDIES

We perform ablation studies on *CodeChemist* to analyze its key components, focusing on the contributions of the multi-temperature hedged sampling and test case generation strategies.

**Sampling Strategy.** We use Pass@1 to measure the diversity of candidate pools and compare two schemes: (i) generating 10 samples with a fixed temperature of $\tau = 0.7$ (following the S* setting), $\tau = 0.9$, and $\tau = 1.1$; (ii) multi-temperature hedged sampling, which generates 1, 3, 3, and 3 samples at $\tau = 0, 0.7, 0.9, 1.1$, respectively. The results are shown in Table 4. The experiments indicate that hedged sampling outperforms single-temperature sampling in most cases, highlighting the importance of balancing stability and diversity through the multi-temperature setting.

**Test Case Generation.** In test case generation, we produce 10 samples from high-resource PLs for voting to create test oracles. The number of generated samples of high-resource PLs can affect the accuracy of these test oracles, which in turn impacts generation performance in low-resource PLs. To validate this, we compare the performance with that obtained using only a single sample from high-resource PLs for test oracle generation. Results in Table 5 show that the voting strategy consistently outperforms the single-sample baseline across all PLs and models, with particularly pronounced gains for smaller models. This is because single decoding from smaller models is more

Table 3: Pass@1 results on Ag-LiveCodeBench-X. Green arrows and values indicate improvements over the vanilla baseline, while red arrows and values denote a decrease in performance.

| Language | Method | Qwen 2.5 Coder Instruct | | | | Llama3 3B | GPT 4o -mini | DeepSeek V3.1 |
|---|---|---|---|---|---|---|---|---|
| | | 1.5B | 7B | 14B | 32B | | | |
| **Lua** | Vanilla | 2.8 | 6.7 | 10.4 | 21.5 | 1.8 | 24.5 | 36.5 |
| | Ours | 6.8 ↑ 4.0 | 17.6 ↑ 10.9 | 26.1 ↑ 15.7 | 33.1 ↑ 11.6 | 5.4 ↑ 3.6 | 28.9 ↑ 4.4 | 50.5 ↑ 14.0 |
| **C++** | Vanilla | 8.4 | 18.8 | 31.6 | 36.8 | 8.6 | 36.7 | 65.0 |
| | Ours | 11.2 ↑ 2.8 | 25.9 ↑ 7.1 | 33.9 ↑ 2.3 | 40.9 ↑ 4.1 | 13.4 ↑ 4.8 | 40.7 ↑ 4.0 | 72.3 ↑ 7.3 |
| **Java** | Vanilla | 5.2 | 11.8 | 31.3 | 28.4 | 5.6 | 37.1 | 61.4 |
| | Ours | 7.4 ↑ 2.2 | 14.4 ↑ 2.6 | 35.7 ↑ 4.4 | 42.1 ↑ 13.7 | 11.6 ↑ 6.0 | 39.3 ↑ 2.2 | 71.3 ↑ 9.9 |

Table 4: Comparing Pass@1 Scores: Single-Temperature Sampling (STS) vs. Multi-Temperature Hedged Sampling (MTHS)

| Sampling | Qwen 3B | | | | Qwen 32B | | | |
|---|---|---|---|---|---|---|---|---|
| | Lua | C++ | Java | Val | Lua | C++ | Java | Val |
| STS(0.7) | 57.6 | 63.2 | 56.3 | 59.0 | 77.1 | **84.2** | 79.3 | 80.2 |
| STS(0.9) | 56.5 | **64.0** | 55.1 | 58.7 | 77.1 | 83.9 | 80.7 | 80.6 |
| STS(1.1) | 55.3 | 61.5 | 48.7 | 55.2 | 76.4 | 82.2 | **84.2** | 80.9 |
| MTHS | **58.2** | 63.9 | **58.0** | **60.0** | **78.0** | 83.9 | 81.5 | **81.1** |

Table 5: Comparing Pass@1 Scores of *Code-Chemist*: Single vs. Ten Candidates from High-Resource PLs

| #Candidates | Qwen 3B | | | Qwen 32B | | |
|---|---|---|---|---|---|---|
| | Lua | C++ | Java | Lua | C++ | Java |
| One Candidate | 75.2 | 71.4 | 82.3 | 82.0 | 86.3 | 87.3 |
| Ten Candidates | **77.6** | **73.3** | **83.5** | **82.6** | **87.0** | **88.6** |

prone to randomness and higher error rates, leading to larger output variance for the same input, while multi-sample voting effectively suppresses hallucinations and incidental errors.

## 4.5 DISCUSSION

Here, we discuss the time cost of *CodeChemist* compared with other test time scaling methods, as well as the potential for combining *CodeChemist* with these methods.

**Time Cost.** Since *CodeChemist* first generates high-resource PL outputs, it introduces additional time cost. To quantify this, we compare the runtime of different methods on Qwen-3B, as shown in Table 7. The results show that *CodeChemist* incurs a higher time cost than LLM Judge and the Majority Voting baseline, but remains substantially lower than S*, which is the second-best method in terms of performance. For instance, on average, the time cost of S* is 5.65× that of *CodeChemist*, while its performance is consistently inferior to ours.

**Token Consumption.** We also compare the token consumption of different methods on Qwen-3B, as shown in Table 6. CodeChemist incurs a slightly higher token consumption compared to the LLM Judge baseline, but remains significantly lower than S*. The primary reason is that S* relies on iterative LLM calls, repeatedly generating adaptive test inputs and performing pairwise comparisons among candidate solutions. In contrast, CodeChemist's token usage is dominated by a one-time generation of reference code and test cases, resulting in a more controlled and efficient overall cost.

Furthermore, we investigate ways to further reduce the inference cost of high-resource PL generation. Specifically, for the Qwen2.5-Coder-Instruct 32B model, instead of using the model itself to generate high-resource PL candidates, we first use the faster Qwen2.5-Coder-Instruct 3B model, although this sacrifices some quality in the high-resource PL generation. Compared to generating high-resource PL candidates with the 32B model, which improved Lua performance from 79.5 (Vanilla) to 82.6, using the 3B model to generate high-resource PL candidates increases the 32B model's performance on Lua to 81.4. Moreover, the time cost decreases from 31.67s to 22.03s. Overall, these results confirm that *CodeChemist* could balance performance gains and computational efficiency, making it a practical solution for enhancing low-resource PL performance.

**Combination with Other Test Time Scaling Methods.** *CodeChemist* can serve as a foundational framework that can be combined with existing test-time scaling methods. The core of *CodeChemist* lies in achieving cross-language knowledge transfer through test case generation, whereas existing methods, such as S*, primarily focus on optimizing candidate selection within a single language.

Table 6: Token usage comparison per problem.

| Method | Lua | C++ | Java | Avg. |
|---|---|---|---|---|
| Vanilla | 328.12 | 413.56 | 652.87 | 464.85 |
| Maj Voting | 2922.12 | 3272.13 | 4796.88 | 3663.71 |
| LLM Judge | 3915.52 | 5005.22 | 7027.74 | 5316.16 |
| S* | 6266.99 | 10324.75 | 11671.25 | 9421.00 |
| Ours | 5199.32 | 5549.33 | 7074.07 | 5940.91 |

Table 7: Time cost comparison on Qwen 3B (seconds/problem).

| Method | Lua | C++ | Java | Avg. |
|---|---|---|---|---|
| Vanilla | 0.65 | 2.49 | 0.85 | 1.33 |
| LLM Judge | 16.58 | 13.03 | 16.93 | 15.51 |
| Majority Vote | 6.34 | 43.50 | 21.08 | 23.64 |
| S* | 52.46 | 232.03 | 203.86 | 162.78 |
| Ours | 19.00 | 40.56 | 24.86 | 28.81 |

Therefore, we explore the effectiveness of the combination of *CodeChemist* and S* on the MultiPL-HumanEval dataset (details are provided in the Appendix E). The combination achieves a Pass@1 score of 83.9 on Qwen 7B on Lua, surpassing both *CodeChemist* (82.0) and S* (79.5). The results highlight the strong compatibility of *CodeChemist* with the other test-time scaling method and its ability to be seamlessly integrated to produce additional gains.

**Future Work.** CodeChemist can further extend to more complex scenarios, such as function state mutations and cross-function interactions. A promising direction is to incorporate cross-language serialization standards (e.g., JSON) to convert internal program states into language-agnostic representations.

## 5 CONCLUSION

We propose *CodeChemist*, a novel test-time scaling framework that transfers functional knowledge from high-resource PLs to low-resource PLs through synthesized test cases. By generating and executing test inputs in high-resource PLs to capture expected behavior, and then leveraging multi-temperature hedged sampling to produce candidate code snippets in the target low-resource language, *CodeChemist* effectively enhances code generation performance. *CodeChemist* is applicable to a wide range of scenarios, such as natural language driven code generation. Extensive experiments on MultiPL-E and Ag-LiveCodeBench-X demonstrate that *CodeChemist* consistently improves performance for low-resource PLs, especially when the capability gap between high- and low-resource languages is large. Results show that *CodeChemist* outperforms existing test-time scaling methods across multiple benchmarks, achieving significant and stable gains.

ETHICS STATEMENT

This work presents a method for low-resource code generation and does not raise any apparent ethical issues. Our research involves automated code generation from natural language or structured specifications, and does not involve human subjects, sensitive data, or discriminatory applications. The datasets used in this study are publicly available and do not contain personal information. We have reviewed the ICLR Code of Ethics and confirm that this work complies with its guidelines. There are no potential conflicts of interest to declare.

REPRODUCIBILITY STATEMENT

To support the reproducibility of our work, we have taken the following steps: We provide an anonymized code repository as supplemental material, which includes the implementation of *CodeChemist* and the related scripts. Detailed descriptions of the experimental setup, including hyperparameters and environment configuration. For the datasets used in our experiments, we provide comprehensive preprocessing steps and relevant download links in the supplemental materials. We hope these resources will facilitate the replication of our results. The repository can be accessed at: (https://anonymous.4open.science/r/CodeChemist-4379).

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

## A  LLM Usage Statement

In the preparation of this paper, we utilized large language models (specifically GPT-4) as an assistive tool for writing and editing purposes. The model was used to help improve the clarity and fluency of certain textual passages, and to assist in polishing the manuscript's language. However, all technical content, research ideas, methodological development, experimental design, result analysis, and scientific conclusions remain the original work of the authors. The authors take full responsibility for the entire content of this paper, including any parts that were edited with LLM assistance. The LLM was not involved in the research ideation process and is not considered a contributor to the intellectual contributions of this work.

## B  Hypothesis Testing

We demonstrate the statistical significance of our results through t-tests. Specifically, we conduct t-tests on the performance results of Qwen1.5B, Qwen7B, Qwen14B, and DeepSeek-V3.1, running the experiments five times with different random seeds. We find that for all these settings, the p-values were $< 0.05$.

## C  Supplemental Experiments

### C.1  Extended Comparisons with Other Methods

Table 8: Pass@1 of Vanilla, Self-Debugging and CodeChemist on MultiPL-HumanEval.

| Method | Qwen 1.5B | | | Qwen 32B | | | DeepSeek V3.1 | | |
|---|---|---|---|---|---|---|---|---|---|
| | Lua | C++ | Java | Lua | C++ | Java | Lua | C++ | Java |
| Vanilla | 34.1 | 34.4 | 43.5 | 78.0 | 83.9 | 81.5 | 82.1 | 93.0 | 89.3 |
| Self-Debugging | 47.8 | 49.7 | 61.4 | 81.4 | 85.1 | 84.2 | 85.7 | 93.2 | 91.1 |
| **Ours** | **57.8** | **52.2** | **69.6** | **82.6** | **87.0** | **88.6** | **88.2** | **95.0** | **93.7** |

We further introduce Self-Debugging as a comparison baseline. we use test oracles generated by 4o-mini for iterative debugging. The revision process stops either when the number of iterations exceeds two or when all test cases are passed. As shown in Table 8, on the MultiPL-HumanEval benchmark, Self-Debugging relies on an external LLM to generate test oracles and perform iterative correction. This allows it to clearly outperform Vanilla. However, CodeChemist achieves even better performance while using no external LLM. The gain of Self-Debugging comes from the combination of external test oracles and execution feedback, whereas the additional gain of CodeChemist comes from its cross PL function transfer mechanism. When a model faces a cognitive bottleneck due to scarce data in a low resource PL, error feedback within the same PL is often unable to correct the underlying misunderstanding. In contrast, CodeChemist uses functional specifications extracted from a high resource PL to build high quality test cases for the low resource language, which leads to a more fundamental improvement in capability.

### C.2  Extended Comparisons On Other Benchmarks

We further conducted comprehensive comparisons against existing methods (Majority Voting and LLM Judge) on MultiPL-MBPP and Ag-LiveCodeBench-X, as show in Table 9. The results show that CodeChemist consistently outperforms all baselines across every PL and model, indicating that its advantages are not limited to MultiPL-HumanEval but also hold robustly on more complex and diverse benchmarks.

Table 9: Comparison with existing methods on the MultiPL-MBPP and Ag-LiveCodeBench-X benchmarks.

| Language | Method | MultiPL-MBPP | | | Ag-LiveCodeBench-X | | |
| | | Qwen (1.5B) | Qwen (32B) | DeepseekV3.1 | Qwen (1.5B) | Qwen (32B) | DeepseekV3.1 |
|---|---|---|---|---|---|---|---|
| **Lua** | Vanilla | 36.85 | 60.71 | 59.45 | 2.81 | 21.52 | 36.53 |
| | Maj Voting | 55.92 (51.75%) | 63.22 (4.13%) | 64.23 (11.11%) | 6.21 (121.0%) | 30.66 (42.47%) | 47.29 (29.46%) |
| | LLM Judge | 49.12 (33.3%) | 60.96 (0.41%) | 55.67 (3.7%) | 4.21 (49.82%) | 23.85 (10.83%) | 38.88 (6.43%) |
| | **Ours** | **57.93 (57.2%)** | **66.5 (9.54%)** | **65.49 (13.28%)** | **6.81 (142.35%)** | **33.07 (53.67%)** | **50.5 (38.24%)** |
| **C++** | Vanilla | 39.6 | 65.84 | 62.64 | 8.38 | 36.75 | 64.99 |
| | Maj Voting | 53.15 (34.22%) | 65.74 (-0.15%) | 68.77 (9.79%) | **11.22 (33.89%)** | 38.9 (5.8%) | 70.34 (8.23%) |
| | LLM Judge | 55.42 (39.95%) | 64.23 (-2.45%) | 68.01 (8.57%) | 10.42 (24.34%) | 38.08 (3.62%) | 67.74 (4.23%) |
| | **Ours** | **56.93 (43.76%)** | **68.51 (4.06%)** | **71.03 (13.39%)** | **11.22 (33.89%)** | **40.88 (11.24%)** | **72.34 (11.31%)** |
| **Java** | Vanilla | 44.72 | 62.62 | 66.11 | 5.17 | 28.44 | 61.38 |
| | Maj Voting | 53.37 (19.34%) | 66.84 (6.74%) | 69.43 (5.02%) | 7.21 (39.46%) | **42.08 (47.96%)** | 67.94 (10.69%) |
| | LLM Judge | 51.3 (14.71%) | 62.69 (0.11%) | 63.47 (3.99%) | 7.01 (35.59%) | 32.46 (14.14%) | 65.13 (6.11%) |
| | **Ours** | **57.77 (29.18%)** | **67.88 (8.4%)** | **71.24 (7.76%)** | **7.41 (43.33%)** | **42.08 (47.96%)** | **71.34 (16.23%)** |

## C.3 A Case Study of C++ to Rust

Table 10: Pass@1 results when using C++ to improve Rust code generation on MultiPL-HumanEval.

| Model | Vanilla | CodeChemist | Rel. Gain (%) |
|---|---|---|---|
| Qwen-7B | 74.17 | 80.77 | +8.9% |
| DeepSeek-V3.2 | 89.74 | 92.31 | +2.9% |

CodeChemist does not rely on Python as a specific source PL. To verify this, we conducted an additional experiment in which C++ serves as the teacher PL to enhance Rust code generation, as show in Table 10. The results on MultiPL-HumanEval show that CodeChemist remains effective in this setting: the performance of Qwen-7B improves from 74.17 to 80.77 (+8.9%), and DeepSeek-V3.2 improves from 89.74 to 92.31 (+2.9%). These results provide initial evidence for the generality of the framework.

## C.4 A Case of Thinking Mode in DeepSeek

Table 11: A Case of Thinking Mode in DeepSeek on MultiPL-HumanEval

| Language | Vanilla | CodeChemist | Rel. Gain (%) |
|---|---|---|---|
| C++ | 92.73 | 94.41 | +1.8% |
| Java | 92.53 | 93.67 | +1.2% |
| Lua | 84.60 | 88.82 | +5.0% |

We evaluate the thinking mode of DeepSeekV3.2 on the MultiPL-HumanEval benchmark, as show in Table 11.. The results show consistent improvements across programming languages, with scores increasing from 92.73 to 94.41 for C++, from 92.53 to 93.67 for Java, and from 84.60 to 88.82 for Lua. These findings demonstrate that CodeChemist remains effective under the thinking mode setting.

## C.5 Hyperparameter Analysis

**Analysis of the Number of Candidate Solutions.** Figure 2 shows the Pass@k scores (vertical axis) for the Qwen 7B model as the number of candidate solutions increases. Based on the observation that the performance gains for all PLs diminish beyond 9-10 candidates, we empirically determined 10 to be the optimal number, balancing computational cost and performance.

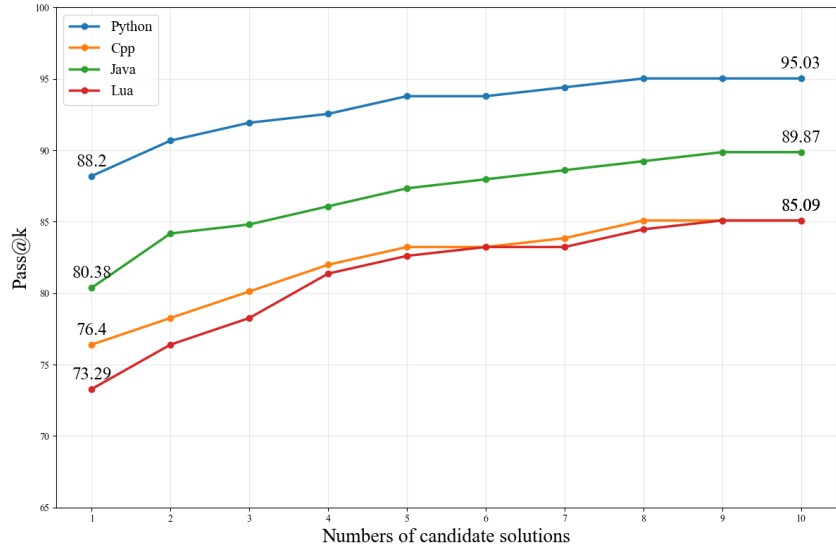

Figure 2: Analysis of the Number of Candidate Solutions On Qwen 7B

## D    PERFORMANCE OF PYTHON VANILLA INFERENCE

As shown in Table 12, the table presents the Pass@1 results of Python native inference across three widely used benchmarks: MultiPL-HumanEval, MultiPL-MBPP, and Ag-LiveCodeBench-X. These results highlight the performance of different models, providing a basis for comparative analysis during the knowledge transfer process in *CodeChemist*.

Table 12: Pass@1 results of Python Vanilla inference performance across three benchmarks: MultiPL-HumanEval, MultiPL-MBPP, and Ag-LiveCodeBench-X.

| Benchmark | Qwen 2.5 Coder Instruct | | | | Llama3 3B | GPT 4o -mini | DeepSeek V3.1 |
|---|---|---|---|---|---|---|---|
| | 1.5B | 7B | 14B | 32B | | | |
| MultiPL-HumanEval | 63.9 | 87.6 | 89.2 | 91.9 | 57.8 | 87.7 | 93.0 |
| MultiPL-MBPP | 45.8 | 70.4 | 72.2 | 76.5 | 55.2 | 70.2 | 78.3 |
| Ag-LiveCodeBench-X | 7.7 | 20.9 | 31.7 | 36.4 | 17.0 | 50.0 | 70.1 |

## E    COMBINATION WITH OTHER TEST-TIME SCALING METHODS

Zhuang et al. (2025) In this section, we explore how *CodeChemist* can be integrated with existing test-time scaling methods to enhance their performance. *CodeChemist* serves as a foundational framework designed to facilitate cross-language knowledge transfer through test case generation, which is distinct from traditional test-time scaling methods. While *CodeChemist*'s core focus is on generating high-quality, language-agnostic test cases to improve the performance of low-resource programming languages, other test-time scaling methods, such as S*, primarily concentrate on optimizing the candidate selection process within a single language.

We explore the combination of *CodeChemist* and the S* method. First, we generate a sample pool using high-temperature hedged sampling, and use Python to create language-agnostic test cases, followed by an initial filtering of the sample pool. Next, we compare the filtered samples pairwise, using LLM to generate inputs that can effectively distinguish between the two solutions. Then, we execute these adaptive inputs and provide feedback to the LLM based on the output, guiding it to make the optimal choice. In the Lua language experiment conducted on Qwen 2.5 Coder Instruct

7B, the performance improved from 69.7 to 83.9, further validating that *CodeChemist* can effectively combine with S* and significantly enhance the code generation capability for low-resource PLs.

## F PROMPTS

In this appendix, we provide the detailed prompts used in our experiments. Our prompts are categorized by benchmark and by task type: (1) code generation and (2) test case generation. For reproducibility, we present the model's prompts.

### F.1 MULTIPL-E

#### F.1.1 CODE GENERATION

---
**Example code generation prompt**

**Prompt:** Please continue to complete the function and return all completed code in a codeblock. Here is the given code to do completion:
```
Question:{}
```
---

#### F.1.2 TEST CASE GENERATION

---
**Example Test Case Generation prompt**

**Prompt:** Please generate 10 diverse and meaningful test case inputs that thoroughly evaluate different aspects of the problem. Insert your test case inputs in the parentheses below and return only the code block:
Question: {}
```
# YOUR test case input HERE#
```
---

### F.2 AG-LIVECODEBENCH-X

#### F.2.1 CODE GENERATION

---
**Example Code Generation prompt**

**Prompt:** You are a helpful assistant. You will be given a question (problem specification) and will generate a correct language program that matches the specification and passes all tests. You will NOT return anything except for the program.
Question: {}
Read the inputs from stdin solve the problem and write the answer to stdout (do not directly test on the sample inputs). Enclose your code within delimiters as follows.
```
# YOUR CODE HERE#
```
---

### F.2.2 TEST CASE GENERATION

---

**Example Test Case Generation prompt**

**Prompt:** You will be given a question (problem specification) and will generate 10 diverse and meaningful test case inputs that thoroughly evaluate different aspects of the question.
Problem: {}
Please read the input format carefully, directly return the generated test case, and do not generate code.
```
# YOUR test case input HERE#
```

---

