# OpenReview forum: "CodeChemist: Test-Time Scaling for Low-Resource Code Generation via Functional Knowledge Transfer"
_ICLR.cc/2026/Conference — Submitted to ICLR 2026_

### Official Review · Reviewer_a8wg · 2025-10-27

**Soundness:** 2
**Presentation:** 3
**Contribution:** 2
**Rating:** 4
**Confidence:** 4

**Summary:**

This paper introduces CodeChemist, a test-time framework that transfers functional knowledge from high-resource programming languages to low-resource ones by synthesizing executable, language-agnostic test cases; it then applies hedged multi-temperature sampling in the target language and selects the candidate maximizing test pass rate—without retraining. Experiments across multiple models and benchmarks report consistent gains and show compatibility with other test-time scaling methods.

**Strengths:**

1. The paper is clearly written and easy to follow.
2. The method is training-free,and the efficiency study indicates that its runtime is acceptable.

**Weaknesses:**

1. The proposed approach appears to primarily compose existing test-time techniques, without introducing a clearly novel algorithmic mechanism or theoretical insight.
2. While the motivation for the proposed multi-temperature hedged sampling is understandable, the rationale for the specific hyperparameter choices is not sufficiently justified.
3. In practice, inputs differ substantially across programming languages—for example, when a function expects a struct—so how should such inputs be converted across languages?
4. In many practical settings, a Python implementation is infeasible, necessitating the use of other languages; conversely, if the task can already be solved in Python, the added value of generating code in another language is unclear.
5. In your paper, you have characterized C++ as a low-resource language. Given its many application scenarios, such as its translation to Rust, we would be more interested in seeing your method applied to a C++-to-Rust context, which represents a more common real-world application.
6. The improvements appear modest, and the set of comparative baselines is insufficient.
7. The quality of the generated test cases is questionable.

**Questions:**

Please see Weaknesses.

---

> ### Author Response · Authors · 2025-11-26
> **Response to a8wg**
>
> We appreciate the reviewer’s positive comments. The reviewer’s suggestions have provided clear direction for improving the paper, and we will address them one by one.
>
> > Q1：The proposed approach appears to primarily compose existing test-time techniques, without introducing a clearly novel algorithmic mechanism or theoretical insight.
>
> A1: We respectfully disagree with the assessment and argue that the primary novelty of our work lies not in a single algorithmic breakthrough, but in the novel integration and orchestration of existing test-time techniques into a cohesive, goal-driven framework.
> Current test time techniques mainly focus on code generation in a single language setting, especially Python. CodeChemist is the first to apply Best of N to low-resource programming languages and introduces a training free cross language transfer of functional knowledge. At the same time, experiments on several models and benchmarks show that our method is effective and greatly improves the usability of LLMs in other programming languages.
>
>
>
> > Q2：While the motivation for the proposed multi-temperature hedged sampling is understandable, the rationale for the specific hyperparameter choices is not sufficiently justified.
>
> A2: Thank you for pointing out this important issue. The hyperparameter choices for multi-temperature hedging sampling were determined based on preliminary experiments: performance drops significantly when the temperature exceeds 1.2, 0.7 is a common setting (e.g., in S*), and 0.9 and 1.1 are used to enhance the model's exploration ability. We further conduct additional experiments, as shown in the table below.
> The corresponding content has been added to the yellow-highlighted text on page 9, line 451.
>
>
>
>
> |              | Qwen(3B) |          |          |          | Qwen(32B) |          |          |          |
> | ------------ | -------- | -------- | -------- | -------- | --------- | -------- | -------- | -------- |
> | Sampling     | Lua      | C++      | Java     | val      | Lua       | C++      | Java     | val      |
> | STS（T=0.7） | 57.6     | 63.2     | 56.3     | 59.0     | 77.1      | **84.2** | 79.3     | 80.2     |
> | STS（T=0.9） | 56.5     | **64.0** | 55.1     | 58.7     | 77.1      | 83.9     | 80.7     | 80.6     |
> | STS（T=1.1） | 55.3     | 61.5     | 48.7     | 55.2     | 76.4      | 82.2     | **84.2** | 80.9     |
> | MTHS         | **58.2** | 63.9     | **58.0** | **60.0** | **78.0**  | 83.9     | 81.5     | **81.1** |
>
>
>
>
>
>
> > Q3：In practice, inputs differ substantially across programming languages—for example, when a function expects a struct—so how should such inputs be converted across languages?
>
>
> In the Ag-LiveCodeBench-X benchmark, CodeChemist has already addressed the handling of C++ struct types. Specifically, we serialize the struct into a general string format, similar to the format used in online judge systems. We then wrap the C++ candidate code with a small adapter that restores the string back to the struct type. For more complex types, CodeChemist can also use established cross language serialization protocols, such as JSON and Protobuf, which are widely used in microservice systems, to support cross language communication.
>
>
> > Q4：In many practical settings, a Python implementation is infeasible, necessitating the use of other languages; conversely, if the task can already be solved in Python, the added value of generating code in another language is unclear.
>
>
> A4: As you suggest in Q5, there is a large and practical space of problems where the core logic is general and independent of the programming language, but the final implementation programming language is fixed due to engineering needs. For example, cross platform development (e.g. iOS and Android), migration of legacy code bases (e.g. C++ to Rust), and development of multi language libraries (e.g. PyTorch, LibTorch, and js-PyTorch). The value of CodeChemist appears in this common setting where the logic is language independent but the engineering must use a specific language. We do not claim that a single language should solve all problems. Instead, we use Python or other high-resource programming languages as the environment in which LLMs perform best to support real coding tasks that appear in other programming languages.
>
>
>
>
>
> > Q5： Given its many application scenarios, such as its translation to Rust, we would be more interested in seeing your method applied to a C++-to-Rust context, which represents a more common real-world application.
>
> A5: We appreciate the reviewer for this excellent and forward-thinking suggestion. Following your recommendation, we further tested using C++ to enhance Rust, and the results showed: Qwen (7B) improved from 74.17 to 80.77（+8.9%）, and DeepSeekV3.2 improved from 89.74 to 92.31（+2.9%）. The corresponding content has been added to the yellow-highlighted text on page 15, line 781.

---

> > ### Author Response · Authors · 2025-11-26
> > **Response to a8wg**
> >
> > > Q6：The improvements appear modest, and the set of comparative baselines is insufficient.
> >
> > A6: We respectfully disagree with the characterization that our improvements are modest and would like to clarify the significance of our results.
> >
> >
> > Regarding the magnitude of improvement, the gains achieved by our method are substantial. On MultiPL-HumanEval, the maximum relative improvements reach 80.6% for Lua, 51.7% for C++, and 60.0% for Java; on MultiPL-MBPP, the corresponding gains are 56.9%, 43.7%, and 44.7%. The improvements on the more challenging Ag-LiveCodeBench-X benchmark are even more pronounced, reaching 200.0% for Lua, 55.8% for C++, and 107.1% for Java.
> > Notably,
> > CodeChemist improves DeepSeek-V3.1's pass rate on Lua to 50.5, surpassing the current leader, Kimi K2 (43.0), on the Ag-LiveCodeBench-X leaderboard.
> >
> >
> > Regarding the adequacy of baselines, test time scaling for code generation is still at an early stage of development [1], and the available baseline methods are limited. To strengthen the comparison, we additionally include Self-Debugging as a baseline.
> > We use test oracles generated by GPT-4o-mini for iterative debugging. The revision process stops either when the number of iterations exceeds two or when all test cases are passed.
> > As shown in the table, although Self-Debugging, by relying on externally generated oracles and iterative correction, significantly outperforms vanilla generation, CodeChemist still achieves superior performance without relying on any external LLM. The corresponding content has been added to the yellow-highlighted text on page 14, line 738.
> >
> > [1]Li D, Cao S, Cao C, et al. S*: Test time scaling for code generation[J]. arXiv preprint arXiv:2502.14382, 2025.
> >
> >
> >
> >
> > | Language | Method         | Qwen（1.5B） | Qwen（32B） | deepseekV3.1 |
> > | -------- | -------------- | ------------ | ----------- | ------------ |
> > | Lua      | Vanilla        | 34.1         | 78.0        | 82.1         |
> > | Lua      | Self-Debugging | 47.8         | 81.4        | 85.7         |
> > | Lua      | **Ours**       | **57.8**     | **82.6**    | **88.2**     |
> > | C++      | Vanilla        | 34.4         | 83.9        | 93.0         |
> > | C++      | Self-Debugging | 49.7         | 85.1        | 93.2         |
> > | C++      | **Ours**       | **52.2**     | **87.0**    | **95.0**     |
> > | Java     | Vanilla        | 43.5         | 81.5        | 89.3         |
> > | Java     | Self-Debugging | 61.4         | 84.2        | 91.1         |
> > | Java     | **Ours**       | **69.6**     | **88.6**    | **93.7**     |
> >
> >
> >
> > > Q7：The quality of the generated test cases is questionable.
> >
> > A7: For LLM-generated code, achieving a test oracle with 100% formal guarantees of correctness is neither practical nor the objective of this work. The test cases in CodeChemist primarily serve to filter candidate programs that exhibit behavior consistent with the high-resource language. The correctness of the test cases comes from the LLM’s ability in the high-resource language, and the voting strategy reduces hallucination in single samples. The effectiveness of the test cases is directly reflected in the actual code selection results: CodeChemist achieves significant performance gains. In addition, Incorrect test cases often come from incorrect Python code. In such cases, the conditional probability that other languages also fail is usually above 90% (as shown in the table), which is consistent with the observation of Zhuang [1]. Therefore, the main bottleneck of CodeChemist lies in the limit of the LLM’s ability, while the test cases merely serve as a medium for knowledge transfer.
> >
> >
> > |          | MultiPL-HumanEval |           | Ag-LiveCodeBench-X |           |
> > | -------- | ----------------- | --------- | ------------------ | --------- |
> > | Language | Qwen(1.5B)        | Qwen(32B) | Qwen(1.5B)         | Qwen(32B) |
> > | Java     | 0.78              | 0.75      | 0.96               | 0.88      |
> > | C++      | 0.91              | 0.67      | 0.94               | 0.91      |
> > | Lua      | 0.91              | 0.92      | 0.97               | 0.97      |
> >
> >
> > [1] Zhuang et al. The rise and down of babel tower: Investigating the evolution process of multilingual code large language model. In ICLR, 2025.
> >
> >
> >
> >
> >
> >
> > We are grateful for the reviewer’s valuable insights and hope that we have adequately responded to all raised points. If any further clarification is needed, please feel free to contact us.

---

> > > ### Comment · Reviewer_a8wg · 2025-11-27
> > >
> > > We appreciate the author's response. We have the following questions:
> > >
> > > 1. What dataset did you use for C++ to Rust translation?
> > > 2. Need clarification of the majority voting to determine the final output.
> > >  2.a) What's the process?
> > >  2.b) How does it related to the 'unique most frequent output?
> > > 3. What's the relationship between 'increasing temperature' and 'covering both common cases and boundary senarios'?
> > >
> > > We are happy to increase the score based on how these questions develop.

---

> > > > ### Author Response · Authors · 2025-11-28
> > > > **#  Response to a8wg**
> > > >
> > > > Thank you very much for your detailed and helpful comments. We address your questions below.
> > > >
> > > >
> > > > > Q1: What dataset did you use for C++ to Rust translation?
> > > >
> > > >
> > > > We use the MultiPL-HumanEval benchmark. This detail has been added at line 774, highlighted in yellow.
> > > >
> > > >
> > > >
> > > > > Q2: Need clarification of the majority voting to determine the final output. 2.a) What's the process? 2.b) How does it related to the 'unique most frequent output?
> > > >
> > > > 2.a Majority voting procedure: For each test input, we execute ten high-resource language (Python) programs to obtain their outputs. We then select the most frequent value among all valid outputs as the final output.
> > > >
> > > > 2.b The “unique most frequent output” refers to the output that appears most often. In rare cases, if multiple outputs tie for the highest frequency, the test input is deemed non-discriminative and discarded.
> > > >
> > > > We have added the details at line 186, highlighted in yellow.
> > > >
> > > >
> > > > >Q3: What's the relationship between 'increasing temperature' and 'covering both common cases and boundary senarios'?
> > > >
> > > >
> > > > Given an initial temperature τ, We guide the LLM to generate test cases that cover common cases and boundary scenarios using the prompt “Please generate 10 diverse and meaningful test case inputs that thoroughly evaluate different aspects of the problem.” (The full prompt is provided in Appendix F, line 920.)
> > > >
> > > > Regarding “increasing temperature”, a single round of generation may not yield a sufficient number of valid test cases. Therefore, the model attempts to generate inputs multiple times. Specifically，after each attempt, we slightly increase the temperature to encourage greater exploration in subsequent attempts. This strategy, which is also adopted by S*, helps produce more diverse test inputs and overall provides better coverage of common cases and boundary scenarios. We have added the details at line 159 and line 189, highlighted in yellow.
> > > >
> > > > We thank you again for your valuable feedback and insights. If you have any further questions, please feel free to let us know.

---

> ### Comment · Reviewer_a8wg · 2025-11-28
>
> Thank you, I will raise my score to 6, when ICLR's bug is fixed.

---

> > ### Author Response · Authors · 2025-11-28
> > **Response to a8wg**
> >
> > Thank you for reconsidering the score. We appreciate the time you have taken to help improve the quality of our manuscript.

---

### Official Review · Reviewer_ERTr · 2025-10-27

**Soundness:** 2
**Presentation:** 3
**Contribution:** 2
**Rating:** 4
**Confidence:** 4

**Summary:**

This paper introduces CodeChemist, a test-time scaling (TTS) framework designed to improve the performance of CodeLLMs on low-resource programming languages (PLs). The authors define high-resource and low-resource PLs based on differences in CodeLLM performance on the same benchmark problems written in different languages. For example, CodeLLMs typically perform well in Python, making it a high-resource language, whereas C++ tends to have poorer results and is thus considered low-resource.

The proposed method proceeds as follows:

1. Generate multiple code solutions in a high-resource PL (e.g., Python).
2. Generate a set of $n$ input test cases.
3. Execute the Python solutions on these inputs to collect the corresponding outputs.
4. Use majority voting across Python solutions to determine the “correct” outputs.
5. Generate multiple candidate solutions in the target low-resource PL (e.g., C++) using hedged sampling (i.e., varying generation temperatures).
6. Execute the C++ solutions on the same $n$ inputs to obtain their outputs.
7. Compare the outputs of the C++ candidates with the “correct” outputs and select the best-performing one.

In their experiments, the authors evaluate CodeChemist on three benchmarks: MultiPL-E (including HumanEval and MBPP tasks) and Ag-LiveCodeBench-X (derived from LiveCodeBench), and cross three low-resource languages: Lua, C++, and Java. They compare CodeChemist against both the baseline model (without TTS) and other TTS methods using seven models in total: four sizes of Qwen 2.5-Coder, Llama 3 3B, GPT-4o-mini, and DeepSeek-V3.1. The results show consistent improvements over the base models across all benchmarks.

**Strengths:**

1. This paper presents a fresh and insightful perspective on test-time scaling for code generation, exploring a fundamental question about the limiting factor in CodeLLMs: is their weakness primarily due to understanding program semantics (the underlying logic) or handling syntax (the language-specific structure)? By disentangling these aspects through cross-language functional transfer, the work provides a novel way to examine how test-time scaling can bridge gaps between high- and low-resource programming languages.

2. CodeChemist proves is shown effective on code contest benchmarks, demonstrating consistent improvements across all evaluated models and benchmarks. The gains are particularly pronounced for smaller models, such as Qwen-2.5-Coder (1.5B) and Llama 3B, indicating that the proposed method not only scales well but also significantly enhances low-resource performance where model capacity is limited.

**Weaknesses:**

1. Scope: This method seems only useful when generating simple and small programs (i.e., coding contests) that run in the command line. They have to be easy to build and must take only one input and produce only one output.
2. Missing details in design: The authors have not explained some important details in the hyperparameters, such as the number of generated solutions in the high-resource PL and the number of test cases. The *Test Case Generation* paragraph in the ablation studies only justifies the need for sampling 10 high-resource PL solutions for voting to create test oracles, instead of explaining why 10 were sampled and how many inputs were generated.
3. The proposed method seems less effective on larger models.
4. Missing evaluation: The other TTS methods are only evaluated on HumanEval but are missing on the other two benchmarks.
5. Before discussing whether this approach works on these benchmarks, let's consider one of the first principles of programming languages: why do we design different programming languages? Programming languages are designed for different purposes, not just because some are easy to write and others are not. The development of different programming languages aims to address different real-world use cases. In many cases, it is not reasonable or practical to generate a Python solution for tasks such as low-level system or memory control, which are typically the use cases of the "low-resource" languages.

**Questions:**

1. Please address the concerns listed in **Weaknesses**.
2. Please explain the design choice of $n$. What is the number of $n$ when generating inputs, and how is this number determined? Intuitively, a larger $n$ would lead to a more precise selection in the final result.
3. What is the token cost of CodeChemist? The authors discuss the time cost of CodeChemist in comparison to other TTS methods and the vanilla LLM in Section 4.5, but the other and perhaps most important aspect of LLM TTS is missing: how many additional tokens are consumed by this method.
4. The authors mention that the experiments are performed on a single A100 GPU. However, the latency of the execution-based selection method also, and perhaps largely, depends on CPU clock speed.
5. Now that reasoning models are widely adopted, it would be valuable if the authors could compare the performance of a model with CodeChemist versus the same model with reasoning (thinking) enabled.

---

> ### Author Response · Authors · 2025-11-26
> **Response to ERTr**
>
> We appreciate the reviewer’s positive comments and valuable suggestions. The reviewer raised several insightful questions regarding our approach, which we will address one by one.
>
> > Q1：Scope: This method seems only useful when generating simple and small programs (i.e., coding contests) that run in the command line. They have to be easy to build and must take only one input and produce only one output.
>
> A1: We respectfully clarify that CodeChemist is by no means limited to handling simple programs. While our current experiments are indeed based on function-level problems (e.g., HumanEval), this is a common and necessary starting point in the code generation domain. Many top-tier conference papers focus exclusively on function-level code generation [1,2,3,4].
>
> Moreover, the Ag-LiveCodeBench-X benchmark we used, which inherits from LiveCodeBench, involves complex data types (e.g., vector, map, struct in C++; Array, Map, Queue in Java; Table in Lua) and frequently requires handling multiple inputs and outputs. In addition, we further explored the potential of applying CodeChemist to more complex scenarios, such as tasks involving state modifications and interactions across functions. As a proof of concept, we implemented a demonstration program and have included it in the supplementary materials.
>
>
> [1] Lavon et al. "Execution Guided Line-by-Line Code Generation." （NeurIPS 2025）\
> [2] Light et al. "Sfs: Smarter code space search improves llm inference scaling." （ICLR 2025）\
> [3] Liu et al. "Revisiting Chain-of-Thought in Code Generation: Do Language Models Need to Learn Reasoning before Coding?."（ICML 2025）\
> [4] Chen et al. "Teaching large language models to self-debug."（ICLR 2024）
>
> > Q2：Missing details in design: The authors have not explained some important details in the hyperparameters, such as the number of generated solutions in the high-resource PL and the number of test cases. The *Test Case Generation* paragraph in the ablation studies only justifies the need for sampling 10 high-resource PL solutions for voting to create test oracles, instead of explaining why 10 were sampled and how many inputs were generated.\
> > Q6：Please explain the design choice of n. What is the number of  when generating inputs, and how is this number determined? Intuitively, a larger  would lead to a more precise selection in the final result.
>
> A2&A6: We thank the reviewer for pointing out this important omission. The number of candidate solutions (N=10) and test cases (M=10) generated in high-resource PLs were determined based on preliminary experiments. For the number of candidate solutions N, we measure the pass@k performance when generating k samples. As shown in Appendix C.5 Figure 2, the performance of all languages becomes saturated once the number of candidate solutions exceeds nine. Similarly, for the number of test cases M, we observed that 8–10 test cases are sufficient to effectively distinguish the majority of candidate solutions, while fewer than 5 lead to inadequate filtering. Therefore, we chose N=10 and M=10 as a reasonable balance between accuracy and efficiency.
> The corresponding content has been added to the yellow-highlighted text on page 15, line 806.
>
>
>
>
> > Q3：The proposed method seems less effective on larger models.
>
> A3: We respectfully disagree with the characterization of our method’s performance on larger models and would like to clarify its true effectiveness.
> The effectiveness of CodeChemist primarily depends on the performance gap between high-resource and low-resource PLs, rather than the scale of the model itself. For example, in DeepSeek-V3.1 on MultiPL-HumanEval, the improvements for Java and C++ are relatively small (+4.9% and +2.2%, respectively). This is partly because the baseline performance of these PLs is already high, and partly because the performance gap between Python and Java, C++ is small. However, in other cases for DeepSeek-V3.1, CodeChemist still achieves significant improvements. Specifically, in MultiPL-HumanEval, the Lua pass rate increases from 82.1 to 88.2 (+7.4%). On MultiPL-MBPP, the improvements for Lua, Java, and C++ are 13.3%, 7.4%, and 13.4%, respectively. On Ag-LiveCodeBench-X, the improvements for Lua, Java, and C++ are 38.4%, 11.2%, and 16.1%.
> Notably,
> CodeChemist improves DeepSeek-V3.1's pass rate on Lua to 50.5, surpassing the current leader, Kimi K2 (43.0), on the Ag-LiveCodeBench-X leaderboard.
>
>
> > Q4：Missing evaluation: The other TTS methods are only evaluated on HumanEval but are missing on the other two benchmarks.
>
> A4: We thank the reviewer for pointing out this oversight. Following the suggestion, we have added a comprehensive comparison with existing methods (Majority Voting, LLM Judge) on MultiPL-MBPP and Ag-LiveCodeBench-X.
> The corresponding content has been added to the yellow-highlighted text in Appendix C.2 (p. 14, line 753)

---

> ### Author Response · Authors · 2025-11-26
> **Response to ERTr**
>
> > Q5：Before discussing whether this approach works on these benchmarks, let's consider one of the first principles of programming languages: why do we design different programming languages? Programming languages are designed for different purposes, not just because some are easy to write and others are not. The development of different programming languages aims to address different real-world use cases. In many cases, it is not reasonable or practical to generate a Python solution for tasks such as low-level system or memory control, which are typically the use cases of the "low-resource" languages.
>
> A5: In fact, there is a large and practical space of problems where the core logic is general and independent of the programming language, but the final implementation programming language is fixed due to engineering needs. The value of CodeChemist appears in this space.
> For example, cross platform development (e.g. iOS and Android), migration of legacy code bases (e.g. C++ to Rust), and development of multi language libraries (e.g. PyTorch, LibTorch, and js-PyTorch).
>
> > Q7：What is the token cost of CodeChemist? The authors discuss the time cost of CodeChemist in comparison to other TTS methods and the vanilla LLM in Section 4.5, but the other and perhaps most important aspect of LLM TTS is missing: how many additional tokens are consumed by this method.
>
>
> A7: We measure the token usage of CodeChemist on the Qwen 3B model, as shown in the table below. The results show that the token usage of CodeChemist is slightly higher than that of LLM Judge but far lower than that of S*. This is because S* depends on iterative LLM calls, which require repeated generation of adaptive test inputs and pairwise comparison of candidate code. In contrast, the token usage of CodeChemist mainly comes from the one time generation of code and test cases, so the total cost is more controlled. To further reduce token usage, CodeChemist supports flexible control of the number of generated samples. In addition, as described in Section 4.5, CodeChemist shows strong cross model scalability. For example, when using a large model, such as 32B, to generate low-resource code, the high-resource code generation task can be moved to a smaller model, such as 3B. This reduces the total token cost while keeping the performance gain.
> The corresponding content has been added to the yellow-highlighted text on page 9, line 466.
>
>
> | Method     | Lua     | C++      | Java     | Average |
> | ---------- | ------- | -------- | -------- | ------- |
> | Vanilla    | 328.12  | 413.56   | 652.87   | 464.852 |
> | Maj Voting | 2922.12 | 3272.13  | 4796.88  | 3663.71 |
> | LLM Judge  | 3915.52 | 5005.22  | 7027.74  | 5316.16 |
> | S*         | 6266.99 | 10324.75 | 11671.25 | 9421.00 |
> | Ours       | 5199.32 | 5549.33  | 7074.07  | 5940.91 |
>
> > Q8：The authors mention that the experiments are performed on a single A100 GPU. However, the latency of the execution-based selection method also, and perhaps largely, depends on CPU clock speed.
>
> A8: All our experiments run on a server with two Intel Xeon Platinum 8163 CPUs, with a total of 48 cores and hyper threading enabled, to ensure a consistent execution environment. We have added this information to the highlighted part on page 7, line 326.
>
>
> > Q9：Now that reasoning models are widely adopted, it would be valuable if the authors could compare the performance of a model with CodeChemist versus the same model with reasoning (thinking) enabled.
>
> A9: Following your suggestion, we test the thinking mode of DeepSeek on MultiPL-HumanEval. The results are: C++ (92.73 to 94.41), Java (92.53 to 93.67), and Lua (84.60 to 88.82). We find that CodeChemist remains effective under the thinking mode.
> The corresponding content has been added to the yellow-highlighted text on page 15, line 799.
>
>
> We once again thank the reviewer for their insightful comments and hope that we have adequately addressed these issues. If there are any further questions, please feel free to let us know.

---

> > ### Comment · Reviewer_ERTr · 2025-11-28
> >
> > I appreciate the authors'  detailed rebuttal. However, some of my concerns remain unsolved.
> >
> > > A1: We respectfully clarify that CodeChemist is by no means limited to handling simple programs. While our current experiments are indeed based on function-level problems (e.g., HumanEval), this is a common and necessary starting point in the code generation domain. Many top-tier conference papers focus exclusively on function-level code generation [1,2,3,4].
> >
> > I agree that function-level benchmarks are a common starting of evaluating code generation methods. However, I do not see how CodeChemist could be applied to real-world tasks like the other work. If the authors wish, please provide some meaningful illustration of the method used for complex software engineering tasks. For example, repo-level function generation/completion with other context dependencies or re-generating a function to fix some issue in the codebase. Experimental results for this are not needed in the rebuttal, but the authors should provide a convincing illustration.
> >
> >  > A5: In fact, there is a large and practical space of problems where the core logic is general and independent of the programming language, but the final implementation programming language is fixed due to engineering needs. The value of CodeChemist appears in this space. For example, cross platform development (e.g. iOS and Android), migration of legacy code bases (e.g. C++ to Rust), and development of multi language libraries (e.g. PyTorch, LibTorch, and js-PyTorch).
> >
> > This does address my concern. It is true that the core logic *might* be independent of the PL. However, choosing different languages is for their PL-specific features. For example, most of Rust's efficiency comes from linear types and "monadic" operations (such as `Iterator`). It is the same for choosing C++: memory-wise operation and zero-cost abstraction. I am not convinced such language-specific features could be well utilized in such a generation framework.

---

> ### Author Response · Authors · 2025-11-30
> **# Response to ERTr**
>
> >The applicability boundaries of CodeChemist
>
>
> Your comments on language-specific features and complex software engineering tasks lead us to clarify the capability boundaries of CodeChemist. We sincerely thank you for these insightful questions and address them from the following perspectives.
>
> 1. For language-specific features in programming languages， we agree that the examples you provide are highly representative and insightful, and we observe that our method supports the specific features shown in your examples. These features appear clearly in the code generated by CodeChemist. For example, in MultiPL-HumanEval we observe Rust linear types (id 15 et al.) and "monadic" operations (id 4 et al.) （the Rust experiments are reported in Appendix C.3). In Ag-LiveCodeBench-X we observe C++ memory-wise operations (id 0, 7 et al.) and zero-cost abstraction (id 4, 6 et al.). We argue that this observation indicates CodeChemist is essentially different from code translation, and its core mechanism is to select candidates code based on language-agnostic functional behavior. In other words, our method focuses on what the code does, while how the code is implemented belongs to the ability of the LLM itself. This is the main advantage of CodeChemist.
>
> 2. For some repo-level function generation/completion tasks that rely on natural language descriptions (such as function comments, task requirements, and user instructions), which represent one of the core application scenarios for code LLMs, CodeChemist proves highly suitable, and its workflow remains consistent with function-level generation tasks.
>
> 3. For tasks such as code completion/completion or bug fixing that require the original program in the target language as input, our method is difficult to apply directly. The main reason is that the effectiveness of CodeChemist depends on the performance gap of the LLM between the high-resource and low-resource languages. When the prompt contains a long segment of the original code, this condition may not fully hold. For example, when we generate a new function or class based on an existing C++ library, the context already contains the C++ library. In this case, it may be more accurate to let the model generate C++ code directly instead of a Python version. To mitigate this limitation, we explore two potential approaches. Both approaches are described below using the same example scenario.
>
>
> * **Introducing Code Translation Techniques**. This strategy leverages code translation tools to first convert the existing source code into the high-resource language. The core workflow is as follows: (1) Translate the existing C++ library into Python. (2) Generate test inputs based on the requirements. (3) Produce a Python-based solution, execute the code, and obtain input-output pairs. (4) Generate multiple C++ code candidates and select them using the test cases.
>
>
>
>
>
> - **Integration of Cross-language Calling Mechanisms**. For scenarios that support cross-language interaction, we introduce tools such as pybind11, Py4J, or gRPC. The core workflow is as follows: (1) wrapping the interfaces of the C++ library into Python-callable modules using pybind11; (2) generating test inputs according to the task requirements; (3) producing and executing Python code to construct input-output test pairs; (4) generating multiple C++ code candidates and select them based on the test cases.
>
>
> In summary, the effectiveness of our approach depends on the performance gap of the LLM between high-resource and low-resource programming languages. In most cases, the LLM performs better in high resource languages. For example, in the common scenario of code generation from natural language descriptions, this mechanism works in a stable way. However, in tasks such as code repair, where the prompt includes the original code in the target language as input, this condition no longer holds, so our approach is difficult to apply directly. To mitigate this limitation, we propose two solutions, whose feasibility has been preliminarily validated through experiments. The relevant code is available in the supplementary file "demo4repo-level".
>
> We sincerely thank you again for your valuable comments, which have helped us clarify the scope of our method’s applicability (the corresponding revisions have been marked yellow in line 516). At the same time, we wish to emphasize that our method is well suited to a broad range of scenarios such as natural language driven code generation, and we believe that this does not weaken the novelty of our approach.

---

### Official Review · Reviewer_pt6D · 2025-10-31

**Soundness:** 2
**Presentation:** 2
**Contribution:** 2
**Rating:** 4
**Confidence:** 4

**Summary:**

This paper proposes Code Chemist, a test-time scaling framework designed to improve the performance of CodeLLMs on low-resource PLs. The core problem addressed is the performance gap where models excel in high-resource PLs (like Python) but falter in low-resource ones (like Lua) due to data scarcity. The authors conduct experiments across multiple models (Qwen, Llama3, GPT-4o mini, DeepSeek) and benchmarks (MultiPL-HumanEval, MultiPL-MBPP, Ag-LiveCodeBench-X), claiming that CodeChemist significantly outperforms baseline and other test-time scaling methods, especially on low-resource PLs.

**Strengths:**

The paper tackles the problem of poor CodeLLM performance on low-resource programming languages.

The method is a test-time-only framework, which does not require costly model fine-tuning.

The evaluation across a wide variety of model families and sizes (Qwen, Llama, GPT-4o mini, DeepSeek) and multiple code benchmarks (MultiPL-HumanEval, MBPP, Ag-LiveCodeBench-X) .

**Weaknesses:**

The paper's main weakness is its limited originality. The method is an assemblage of existing techniques: Best-of-N sampling (re-branded as "hedged sampling"), test-case-based validation, and cross-language transfer. The contribution is an engineering one, not a fundamental research advancement, which overstates its novelty.

**Questions:**

The test oracle generation relies on majority voting from 10 high-resource samples. What analysis was done to ensure these oracles are correct? How does the method handle a systematic model failure, where the majority of the 10 samples are logically incorrect in the same way? Have you analyzed the failure cases where CodeChemist succeeds or fails based on the correctness of this generated "ground truth"?

Can you provide a clear and rigorous justification for categorizing C++ and Java as "low-resource" PLs in the context of "data scarcity"?

How does MTHS compare to a standard, well-tuned Best-of-N sampling baseline (e.g., all 10 samples at an optimal fixed $\tau$)?

---

> ### Author Response · Authors · 2025-11-26
> **Response to pt6D**
>
> We appreciate your positive comments and detailed feedback. We address the raised concerns as follows:
> > W1：The paper's main weakness is its limited originality. The method is an assemblage of existing techniques: Best-of-N sampling (re-branded as "hedged sampling"), test-case-based validation, and cross-language transfer. The contribution is an engineering one, not a fundamental research advancement, which overstates its novelty.
>
> We respectfully disagree that hedged sampling and Best-of-N sampling are regarded as equivalent.
> The "multi-temperature hedging sampling" approach aims to avoid finding the optimal temperature for each combination of dataset, model, and programming language, and our ablation study demonstrates its effectiveness.
>
> Regarding the technical novelty, We argue that the core innovation of our work does not lie in a single algorithmic breakthrough, but in the novel integration and orchestration of existing test-time techniques into a cohesive, goal-driven framework. CodeChemist is the first to apply Best-of-N in low-resource PLs, pioneering a paradigm for cross-lingual functional knowledge transfer without the need for training. Ultimately, CodeChemist delivers significant empirical performance improvements.
> > Q1：The test oracle generation relies on majority voting from 10 high-resource samples. What analysis was done to ensure these oracles are correct? How does the method handle a systematic model failure, where the majority of the 10 samples are logically incorrect in the same way? Have you analyzed the failure cases where CodeChemist succeeds or fails based on the correctness of this generated "ground truth"?
> >
>
> A1: On ensuring the correctness of the oracles: For LLM-generated code, achieving a test oracle with 100% formal guarantees of correctness is neither practical nor the objective of this work.
> The test cases in CodeChemist primarily serve to filter candidate programs that exhibit behavior consistent with the high-resource programming language. Their correctness follows from the LLM’s capability in high-resource programming language. Ultimately, the correctness of this oracle is validated by its practical impact on code selection: CodeChemist achieves significant performance gains.
>
> On handling systematic model failure:
> We appreciate your insightful comments. If most high-resource samples share the same logical error, the majority vote oracle and the resulting selection will be inaccurate. To improve the robustness of the framework in such cases, we plan the following enhancement: when all candidate programs achieve consistently low pass rates, we will cluster the low-resource candidates based on their execution outputs across all test inputs and select the answer from the largest cluster.
>
> Failure analysis:
> Incorrect test cases often come from incorrect Python code. In such cases, the conditional probability that other languages also fail is usually above 90% (as shown in the table), which is consistent with the observation of Zhuang et al. [1]. Therefore, the main bottleneck of CodeChemist lies in the limit of the LLM’s ability, while the test cases merely serve as a medium for knowledge transfer.
>
>
> |          | MultiPL-HumanEval |           | Ag-LiveCodeBench-X |           |
> | -------- | ----------------- | --------- | ------------------ | --------- |
> | Language | Qwen(1.5B)        | Qwen(32B) | Qwen(1.5B)         | Qwen(32B) |
> | Java     | 0.78              | 0.75      | 0.96               | 0.88      |
> | C++      | 0.91              | 0.67      | 0.94               | 0.91      |
> | Lua      | 0.91              | 0.92      | 0.97               | 0.97      |
>
> [1] Zhuang et al. The rise and down of babel tower: Investigating the evolution process of multilingual code large language model. In ICLR, 2025.
>
> > Q2：Can you provide a clear and rigorous justification for categorizing C++ and Java as "low-resource" PLs in the context of "data scarcity"?
>
> A2: We apologize for the confusion. The statement "We evaluate the low-resource PL Lua, C++, and Java" in line 313 was misleading and did not accurately reflect our definition of language resources. This statement contradicts the correct descriptions in lines 69 and 356. In line 69, we explicitly classify Lua as a "low-resource PL" and describe C++ and Java as "considerably less low-resource". In the revised version, we have modified the text at page 6, line 314, highlighted in yellow.

---

> ### Author Response · Authors · 2025-11-26
> **Response to pt6D**
>
> > Q3：How does MTHS compare to a standard, well-tuned Best-of-N sampling baseline (e.g., all 10 samples at an optimal fixed )?
>
> A3: As we have provided in Section 4.4（p. 8, line 418), the main advantage of MTHS is not exceeding the optimal temperature after tuning, but rather its ability to achieve robust performance without the need to find the optimal temperature for each model, dataset, and programming language. We have further validated this through additional experiments, as shown in the table below.
> The relevant content has been updated in the yellow-highlighted text at page 9, line 451.
>
> |              | Qwen(3B) |          |          |          | Qwen(32B) |          |          |          |
> | ------------ | -------- | -------- | -------- | -------- | --------- | -------- | -------- | -------- |
> | Sampling     | Lua      | C++      | Java     | val      | Lua       | C++      | Java     | val      |
> | STS（T=0.7） | 57.6     | 63.2     | 56.3     | 59.0     | 77.1      | **84.2** | 79.3     | 80.2     |
> | STS（T=0.9） | 56.5     | **64.0** | 55.1     | 58.7     | 77.1      | 83.9     | 80.7     | 80.6     |
> | STS（T=1.1） | 55.3     | 61.5     | 48.7     | 55.2     | 76.4      | 82.2     | **84.2** | 80.9     |
> | MTHS         | **58.2** | 63.9     | **58.0** | **60.0** | **78.0**  | 83.9     | 81.5     | **81.1** |

---

### Official Review · Reviewer_ftNS · 2025-10-31

**Soundness:** 2
**Presentation:** 3
**Contribution:** 2
**Rating:** 4
**Confidence:** 4

**Summary:**

This paper presents CodeChemist, a test-time framework intended to improve code generation for low-resource programming languages. The core idea is to transfer "functional knowledge" from a high-resource language (like Python) to a low-resource one without any model retraining. The method uses a high-resource language (Python) to first generate code and derive a set of input-output test cases. These test cases are then used as a filter to select the best solution from multiple candidates generated in the target low-resource language. The authors conduct extensive experiments on benchmarks like MultiPL-E and Ag-LiveCodeBench-X, demonstrating that CodeChemist significantly outperforms vanilla generation and other test-time scaling baselines

**Strengths:**

1. The paper tackles the important problem of improving code generation for less-represented languages, which is crucial for the broader applicability of these models.
2. The paper is well-written, and the proposed CodeChemist framework is explained with great clarity. The three-stage process is logical and easy to comprehend, making the work highly accessible.

**Weaknesses:**

1. Inappropriate baseline. The proposed method relies heavily on an external code execution environment to generate and validate solutions. However, the chosen baselines do not leverage this execution feedback in the same way. A more direct comparison would be against methods that also utilize execution feedback. For instance, a baseline could involve an iterative refinement loop where the model corrects its own code in the low-resource language based on feedback from failing test cases. Without such a baseline, it is difficult to determine whether the performance gain truly stems from the cross-language knowledge transfer from Python, or simply from the powerful signal provided by the test cases and the execution environment itself.
2. The idea of using test cases to enhance code generation is a well-established practice in the field. (e.g., GenX, RepoCoder, Epicoder) use test execution to filter, validate, or refine generated code. The authors should clarify the key differences between CodeChemist and these works in the paper.
3. The method for generating test oracles has potential weaknesses. It relies on an LLM to generate inputs and uses majority voting over LLM-generated high-resource code snippets to determine the correct output. This process lacks formal guarantees of correctness or coverage.

**Questions:**

1. How does the framework ensure that the test cases generated via the high-resource PL are sufficient to cover corner cases and complex logic?
2.  The experiments exclusively use Python as the high-resource "teacher" language. How dependent is CodeChemist's success on this specific choice? Have you experimented with other high-resource languages?
3. How do you envision CodeChemist scaling to more complex, multi-function, or modular code generation tasks where correctness depends not just on input-output behavior but also on state modifications and inter-function interactions?

---

> ### Author Response · Authors · 2025-11-26
> **Response to ftNS**
>
> Thank you for your positive comments and detailed feedback. We are glad that you consider our paper to be of significant importance. We will address the insightful questions you raised regarding our approach, one by one.
>
> > Q1：Inappropriate baseline. The proposed method relies heavily on an external code execution environment to generate and validate solutions. However, the chosen baselines do not leverage this execution feedback in the same way. A more direct comparison would be against methods that also utilize execution feedback. For instance, a baseline could involve an iterative refinement loop where the model corrects its own code in the low-resource PL based on feedback from failing test cases. Without such a baseline, it is difficult to determine whether the performance gain truly stems from the cross-language knowledge transfer from Python, or simply from the powerful signal provided by the test cases and the execution environment itself.
>
> A1:
> Regarding the baseline issue raised, the S* method already incorporates an iterative debugging mechanism.
> To migitigate your concern, we further introduce Self-Debugging as a comparison baseline. We use test oracles generated by GPT-4o-mini for iterative debugging.
> As shown in the table below, in the MultiPL-HumanEval benchmark, although the Self-Debugging baseline significantly outperforms the Vanilla method through externally generated test predictions and iterative correction, CodeChemist still demonstrates superior performance without relying on any external LLMs.
> The corresponding content has been added to the yellow-highlighted text on page 14, line 738.
>
>
> | Language | Method         | Qwen（1.5B） | Qwen（32B） | deepseekV3.1 |
> | -------- | -------------- | ------------ | ----------- | ------------ |
> | Lua      | Vanilla        | 34.1         | 78.0        | 82.1         |
> | Lua      | Self-Debugging | 47.8         | 81.4        | 85.7         |
> | Lua      | **Ours**       | **57.8**     | **82.6**    | **88.2**     |
> | C++      | Vanilla        | 34.4         | 83.9        | 93.0         |
> | C++      | Self-Debugging | 49.7         | 85.1        | 93.2         |
> | C++      | **Ours**       | **52.2**     | **87.0**    | **95.0**     |
> | Java     | Vanilla        | 43.5         | 81.5        | 89.3         |
> | Java     | Self-Debugging | 61.4         | 84.2        | 91.1         |
> | Java     | **Ours**       | **69.6**     | **88.6**    | **93.7**     |
>
>
>
>
> > Q2：The idea of using test cases to enhance code generation is a well-established practice in the field. (e.g., GenX, RepoCoder, Epicoder) use test execution to filter, validate, or refine generated code. The authors should clarify the key differences between CodeChemist and these works in the paper.
>
> A2：As described in Section 2.3 (p. 3, line 127), the key difference between CodeChemist and other test case augmented code generation methods is that existing methods focus on code optimization in a single-language environment (especially Python) and rely on the model to directly generate input-output pairs, which often leads to hallucination issues. In contrast, CodeChemist introduces a paradigm for cross-language functional knowledge transfer.
>
> Regarding the work you mentioned, RepoCoder does not use test cases to optimize code generation. Epicoder and GenX still focus on a single-language environment and rely on the LLM to directly generate input-output pairs.
> The corresponding content has been added to the yellow-highlighted text on page 3, line 134.
>
>
> > Q3：The method for generating test oracles has potential weaknesses. It relies on an LLM to generate inputs and uses majority voting over LLM-generated high-resource code snippets to determine the correct output. This process lacks formal guarantees of correctness or coverage.
> >
> > Q4：How does the framework ensure that the test cases generated via the high-resource PL are sufficient to cover corner cases and complex logic?
> >
>
> A3&A4:
> we would like to clarify that the objective of our component is not to construct a comprehensive test suite with full coverage, but to act as a pragmatic filter. The correctness of the test cases comes from the LLM’s ability in the high-resource language, and the voting strategy reduces hallucination in single samples. The effectiveness of the test cases is directly reflected in the actual code selection results: CodeChemist achieves significant performance gains. In addition, Incorrect test cases often come from incorrect Python code. In such cases, the conditional probability that other languages also fail is usually above 90% (as shown in the table), which is consistent with the observation of Zhuang [1]. Therefore, the main bottleneck of CodeChemist lies in the limit of the LLM’s ability, while the test cases merely serve as a medium for knowledge transfer.
>
> [1] Zhuang et al. The rise and down of babel tower: Investigating the evolution process of multilingual code large language model. In ICLR, 2025.

---

> ### Author Response · Authors · 2025-11-26
> **Response to ftNS**
>
> |          | MultiPL-HumanEval |           | Ag-LiveCodeBench-X |           |
> | -------- | ----------------- | --------- | ------------------ | --------- |
> | Language | Qwen(1.5B)        | Qwen(32B) | Qwen(1.5B)         | Qwen(32B) |
> | Java     | 0.78              | 0.75      | 0.96               | 0.88      |
> | C++      | 0.91              | 0.67      | 0.94               | 0.91      |
> | Lua      | 0.91              | 0.92      | 0.97               | 0.97      |
>
>
> > Q5：The experiments exclusively use Python as the high-resource "teacher" language. How dependent is CodeChemist's success on this specific choice? Have you experimented with other high-resource PLs?
>
> A5: CodeChemist does not rely on Python as a "teacher" language. To validate this, we conducted additional experiments using C++ as the "teacher" language to improve Rust code generation. The results on MultiPL-HumanEval show that CodeChemist remains effective in this setting: the performance of the Qwen-7B model improved from 74.17 to 80.77 (+8.9%), and DeepSeek-V3.2 improved from 89.74 to 92.31 (+2.9%). This provides preliminary evidence for the framework's versatility.
> The corresponding content has been added to the yellow-highlighted text on page 15, line 781.
>
>
> > Q6：How do you envision CodeChemist scaling to more complex, multi-function, or modular code generation tasks where correctness depends not just on input-output behavior but also on state modifications and inter-function interactions?
>
>
> A6: We thank the reviewer for pointing out the key directions for our future research.
> For function state modification, we plan to introduce a state serialization interface. This interface uses established cross-language serialization standards, such as JSON and Protocol Buffers, to convert internal states into a language-independent format. By comparing the serialized states, we can achieve cross-language knowledge transfer of program state changes.
> For function interactions, we will guide the LLM to generate a test script that contains a complete sequence of operations and serialize it into a standard format, such as JSON. Then, we will implement parsers in other programming languages to interpret the test script and compare the captured states and return values. We have implemented a simple demo （involving function state modification and function interactions）, which has been uploaded to the supplementary materials.
> The corresponding content has been updated to the yellow-highlighted text on page 10, line 499.
>
>
> Thank you again for your detailed review and constructive suggestions, and we hope we have adequately addressed these issues. Please feel free to let us know if you have any further questions.

---

> > ### Comment · Reviewer_ftNS · 2025-11-28
> >
> > Thank you for the detailed response. The additional experiments and clarifications have effectively addressed most of my concerns. Therefore, I will raise my score to 6.

---

> > > ### Author Response · Authors · 2025-11-28
> > > **Response to ftNS**
> > >
> > > We're glad our rebuttal addressed your concerns. Thank you for your support and encouragement!

---

### Author Response · Authors · 2025-11-26
**Common responses**

We thank all reviewers for the insightful feedback. We have made the following revisions to our paper:

* Line 134: Added the comparison with GenX and Epicoder. (Reviewer ftNS)
* Line 314: Corrected the sentence from "We evaluate the low-resource PL Lua, C++, and Java." to "We evaluate the low-resource PL Lua, along with the relatively low-resource PL C++ and Java." (Reviewer pt6D)
* Line 326: Added server CPU specifications. (Reviewer ERTr)
* Line 450: **Added experiments for temperature selection in multi-temperature hedged sampling.** (Reviewers pt6D, a8wg)
* Line 464: **Added a token consumption analysis with supplementary experiments.** (Reviewer ERTr)
* Line 500: Expanded the discussion of future directions on state modifications and inter-function interactions. (Reviewer ftNS)
* Line 726: **Added an additional comparative baseline.** (Reviewer a8wg, ftNS)
* Line 750: **Added comparative experiments on the MultiPL-MBPP and Ag-LiveCodeBench-X benchmarks.** (Reviewer ERTr)
* Line 774: **Added a case study on C++ to Rust knowledge transfer.** (Reviewers a8wg, ftNS)
* Line 790: **Added a case study of the Thinking Mode in DeepSeek.** (Reviewer ERTr)
* Line 806: Added hyperparameter analysis experiments. (Reviewer ERTr)
* Line 516: **Added the primary applicable scenarios of the method.** (Reviewer ERTr)
* Line 159, 186: Refined the description of the test case generation process. (Reviewer a8wg)


In the revised manuscript, all modifications are highlighted in yellow, and all reviewer comments have been fully addressed in the responses below. We hope that our clarifications adequately resolve all concerns raised by the reviewers. We sincerely appreciate the reviewers’ time and valuable feedback, and we are glad to provide any additional information if needed.

---

### Author Response · Authors · 2025-12-02
**Summary of Discussion**

During the discussion, we engaged in active communication with the reviewers, and most concerns were clarified. **Two reviewers subsequently raised their scores from 4 to 6**. The details are as follows:
- **Reviewer ftNS (4->6, discussion finished)**: We primarily clarified the baseline selection and test oracle generation process, addressing related factual concerns, with the score increased by two points.
- **Reviewer pt6D (4, no discussion)**: In our responses, we primarily clarified the multi-temperature hedged sampling (MTHS) and the test oracle generation process.
- **Reviewer ERTr (4, discussion unfinished):** We primarily clarified the main applicable scenarios and performance of the method. The reviewer further expressed interest in the boundaries of the method's applicability. Since the discussion was closed on the same day, subsequent communication could not continue. We have already submitted a detailed response (noteId: Q7V3kENPfa).
- **Reviewer a8wg (4->6, discussion finished):** We primarily clarified the method novelty, experimental details and test oracle generation, addressing corresponding factual concerns, and the reviewer raised the score by two points.




Meanwhile, we are glad to see that the reviewers have found:
- Our research addresses an important problem (ftNS, pt6D) and presents a fresh and insightful perspective (ERTr).
- Our method is training-free (pt6D, a8wg) and scales well (ERTr).
- Our evaluation is sufficient (pt6D) and demonstrates consistent improvements (ERTr).
- Our paper is well-written (ftNS, a8wg).

We once again sincerely thank all reviewers for their valuable time and constructive comments. This invaluable feedback has not only greatly enhanced the quality of our manuscript but also provided important inspiration for our future research directions.

---

### Meta-Review · Area_Chair_y8d8 · 2026-01-04

**Summary:**

This paper introduces Code Chemist, a test-time scaling method aimed at improving CodeLLM performance on low-resource programming languages. The authors evaluate the approach across several models and benchmarks, reporting consistent gains over baselines and existing test-time scaling methods, particularly for low-resource languages.

However, despite the practical relevance of the problem and the improved task performance, the proposed solution appears narrowly tailored to this specific setting. The methodological contribution is largely incremental (e.g., it is not clear how the proposed method will work in the repo-level setting / in cases were Python can not be used), and the work does not meet the novelty threshold expected for an ICLR submission.

**Reviewer Concerns:**

1. Including additional baselines -> addressed by the authors
2. Writing clarifications -> addressed by the authors
3. Including additional references -> addressed by the authors
4. Evaluation beyond function generation / Python as high-resource PL -> was not addressed by authors
5. Limited novelty -> was not addressed by authors

**Reviewer Scores:**

R. ftNS before: 4 -> after: 6

R. pt6D before: 4 -> after: 4

R. ERTr before: 4 -> after: 4

R. a8wg before: 4 -> after: 6

---

### Decision · Program_Chairs · 2026-01-26

Reject